# NAD$^+$ metabolism is a key modulator of bacterial respiratory epithelial infections

Björn Klabunde[1], André Wesener[1], Wilhelm Bertrams[1], Isabell Beinborn[1], Nicole Paczia[2], Kristin Surmann[3], Sascha Blankenburg[3], Jochen Wilhelm [4,5], Javier Serrania[6], Kèvin Knoops[7], Eslam M. Elsayed[6,8,9], Katrin Laakmann[1], Anna Lena Jung[1,10], Andreas Kirschbaum[11], Sven Hammerschmidt [12], Belal Alshaar[13], Nicolas Gisch [13], Mobarak Abu Mraheil[14], Anke Becker [6], Uwe Völker [3], Evelyn Vollmeister [1], Birke J. Benedikter [1,15,18] ✉ & Bernd Schmeck [1,4,6,10,16,17,18] ✉

Lower respiratory tract infections caused by *Streptococcus pneumoniae* (*Spn*) are a leading cause of death globally. Here we investigate the bronchial epithelial cellular response to *Spn* infection on a transcriptomic, proteomic and metabolic level. We found the NAD$^+$ salvage pathway to be dysregulated upon infection in a cell line model, primary human lung tissue and in vivo in rodents, leading to a reduced production of NAD$^+$. Knockdown of NAD$^+$ salvage enzymes (NAMPT, NMNAT1) increased bacterial replication. NAD$^+$ treatment of *Spn* inhibited its growth while growth of other respiratory pathogens improved. Boosting NAD$^+$ production increased NAD$^+$ levels in immortalized and primary cells and decreased bacterial replication upon infection. NAD$^+$ treatment of *Spn* dysregulated the bacterial metabolism and reduced intrabacterial ATP. Enhancing the bacterial ATP metabolism abolished the antibacterial effect of NAD$^+$. Thus, we identified the NAD$^+$ salvage pathway as an antibacterial pathway in *Spn* infections, predicting an antibacterial mechanism of NAD$^+$.

Lower respiratory infections are the fourth leading cause of death globally, with *Streptococcus pneumoniae* (*Spn*) being the most common respiratory bacterial pathogen[1]. Thus, an in-depth understanding of infection processes on the molecular level is vital to enabling the development of new therapeutic options to face rising antibiotic resistance. Accordingly, host-pathogen interactions during pneumococcal infections are a field of intensive study. However, while most studies have focused on host responses towards *Spn* in dedicated immune cells[2], the respiratory epithelium constitutes the first line of defence against lower respiratory tract infections. It forms a mechanical barrier, produces a thick, glycan-composed mucus to entrap bacteria and secretes various cytokines and chemokines to recruit immune cells to sites of infection. It also promotes direct bacterial killing by secreting antibacterial peptides[3]. Analyzing the immediate

transcriptomic response of infected alveolar epithelial cells up to 4 h post *Spn* infection, Aprianto et al. have demonstrated an extensive dysregulation of gene expression, especially repression of genes involved in the immune response[4]. Yet, the effects of prolonged *Spn* infection on epithelial cell gene expression and metabolism have not yet been investigated.

Here, we employed a multi-omics approach to gain insights into the epithelial response against *Spn*, strain D39, infection. This approach revealed a dysregulation of key enzymes and metabolites of the Nicotinamide adenine dinucleotide (NAD) metabolism. By transitioning between its oxidized (NAD$^+$) and reduced (NADH) forms, NAD acts as key electron transporter in cellular energy pathways. In addition, it is an important cofactor for a wide array of enzymes that convert NAD$^+$ to the energy-depleted metabolite nicotinamide (NAM) (e.g., PARP, sirtuins[5–7]). De novo biosynthesis of NAD$^+$ starts from

diet-derived tryptophan or nicotinic acid[8]. However, the majority of cellular NAD$^+$ is recovered from NAM via the NAD$^+$ salvage pathway[9]. In the first salvage step, nicotinamide phosphoribosyl transferase (NAMPT) catalyzes the production of nicotinamide mononucleotide (NMN) from NAM[10]. Second, nicotinamide-mononucleotide adenylyl transferases (NMNAT) use NMN as a substrate to restore NAD$^+$ [9,11]. The exact contribution of each metabolic pathway to the upkeep of NAD$^+$ homeostasis differs depending on the tissue, environmental conditions and availability of nicotinamide precursors and other nutrients[6,12].

To explore the functional relevance of the altered NAD$^+$ metabolism described here, we manipulated different enzymes or metabolites and explored their effects on *Spn* infection. Thereby, we identified the modulation of the NAD$^+$ metabolism as a previously unknown element in antibacterial defence against *Spn*. Direct treatment of *Spn* with NAD$^+$ inhibited bacterial growth in a concentration-dependent manner, while other bacterial pathogens remained unaffected. Our study deepens the understanding of the interaction between *Spn* and the host metabolism and identifies the NAD$^+$ metabolism as a potential therapeutic target against lower respiratory tract infection.

## Results

### Pneumococcal infection dysregulates the epithelial NAD$^+$ metabolism

To investigate host-pathogen interactions during *Spn* D39 infection of epithelial cells, we established an in vitro infection system using the bronchial epithelial cell line BEAS-2B. To assess the proteomic response of epithelial cells to *Spn* D39 by SILAC proteomics, BEAS-2B cells were infected with MOI 0.5 for 16 h or left uninfected. (Fig. 1A). The 5 most significantly upregulated proteins included two enzymes involved in the NAD$^+$ metabolism, Nicotinamide phosphoribosyl transferase (NAMPT; fold change (FC) = 2.6, $p$ = 0.000002) and Nicotinamide N-Methyl transferase (NNMT; FC = 1.7, $p$ = 0.000148). Significant regulation of the NAD$^+$ metabolism was further confirmed by mRNA microarray data (Fig. 1B). NAMPT and NNMT mRNA were upregulated compared to untreated controls after both *Spn* D39 infection and stimulation with the bacterial cell wall component lipoteichoic acid (LTA) for 9 or 16 h. In addition, mRNA of nicotinamide mononucleotide adenylyl transferase 1 (NMNAT1) was significantly downregulated in BEAS-2B cells exclusively during later-stage infection (16 h). Metabolite measurements by LC-MS/MS revealed that NAD$^+$, as well as multiple NAD$^+$ precursors, were decreased after 16 h of *Spn* D39 infection (Fig. 1C). In summary, we found a clear dysregulation of NAD$^+$ biosynthesis via the Preiss-Handler and NAD$^+$ salvage pathways on a multi-omics scale (Fig. 1D, Supplementary Fig. 1).

We then aimed to confirm the regulation of NAMPT and NMNAT1 upon pneumococcal infection in primary cell culture infection models. Primary human bronchial epithelial cells (hBECs) from healthy donors were differentiated into a pseudostratified lung epithelium at an air-liquid interface. Cells were apically infected with *Spn* D39, MOI 20 for 16 h and gene expression was analyzed by qPCR (Fig. 2A). In infected hBECs, we found expression of NAMPT to be upregulated and NMNAT1 to be downregulated compared to uninfected controls (Fig. 2B, C). In the next step, human lung explants were injected with *Spn* D39 for 12 h and gene expression was determined (Fig. 2D). We found an upregulation of NAMPT and downregulation of NMNAT1, confirming our previous results (Fig. 2E, F). Re-analysis of publicly available gene expression datasets of *Spn* D39 infected mouse lungs equally revealed NAMPT upregulation and NMNAT1 downregulation in vivo. (Fig. 2G, H). In summary, we found the NAD$^+$ salvage gene expression to be dysregulated upon *Spn* D39 infection in primary cell culture and in vivo.

### NAD$^+$ production shows direct antibacterial effects

To assess the role of NAD$^+$ biosynthesis during the infection process, we performed siRNA knockdowns of NAMPT and NMNAT1 48 h prior to infection with *Spn* D39 (MOI 1, 16 h). Successful knockdowns were confirmed on transcript and protein levels (Supplementary Fig. 2). After both knockdowns, the intracellular NAD concentration was significantly decreased, whereas bacterial CFU after 16 h of infection was increased by approximately 40% compared to the scramble control (Fig. 3A, B). Based on these findings, we further addressed whether the antibacterial activity of NAMPT and NMNAT1 is mediated by NAD$^+$ production. The addition of NAD$^+$ to the *Spn* D39 infection of BEAS-2B cells (Fig. 3C; MOI 1, 16 h) or to host cell-free cultures of *Spn* D39 (Fig. 3D–F) revealed a concentration-dependent reduction of bacterial replication, suggesting a direct antibacterial effect of NAD$^+$.

To further assess the role of NAD$^+$ in the infection process, we aimed to increase the host cell NAD$^+$ production during infection. First, we boosted NAD$^+$ production by treatment with the precursor Nicotinamide riboside (NR), which was previously shown to increase intracellular NAD$^+$ in vitro and in vivo[13–15]. NR treatment of BEAS-2B increased the intracellular amount of NAD (NAD$^+$ and NADH) by a factor 2.3, compared to untreated controls (Fig. 4A). In the presence of host cells, NR treatment reduced bacterial replication by factor 0.3 compared to untreated controls (Fig. 4B). The effect could not be detected in the absence of host cells, as NR treatment of *Spn* D39 cultivated in host cell-free medium only caused a minor reduction of bacterial replication by a factor 0.9 (Fig. 4B). Knockdown of NMNAT1 prior to infection and NR treatment reduced NAD production (Fig. 4C) and partially rescued bacterial replication (Fig. 4D). We further confirmed these results using the NAD$^+$ precursor NMN and NAM. In the absence of host cells, both precursors did not affect bacterial replication. In the presence of host cells, NMN and NAM treatment during infection increased intracellular NAD and reduced bacterial replication (Supplementary Fig. 3A–F). Furthermore, chemical inhibition of NAMPT using FK-866 increased bacterial replication (Fig. 4E), while NAMPT activation using SBI-797812 decreased *Spn* D39 replication (Fig. 4F). Next, we validated the effects of NR addition on the course of pneumococcal infection in an air-liquid interface model with well-differentiated primary human bronchial epithelial cells (Fig. 4G–I). During *Spn* D39 infection, basolateral addition of NR caused an increase in intra- and extracellular NAD (Fig. 4G, H) and a reduction of bacterial replication (Fig. 4I). These results were confirmed when using NMN to increase NAD$^+$ in primary human bronchial epithelial cells, cultivated at an air-liquid interface, prior to infection (Supplementary Fig. 3G–I). In summary, NAD$^+$ precursors exhibited no or minor direct antibacterial activity. In contrast, knockdown and chemical modulation of the NAD$^+$ biosynthesis enzymes NMNAT1 and NAMPT, as well as metabolite treatment of *Spn* D39, revealed an anti-pneumococcal effect of NAD$^+$.

### The antibacterial effect of NAD$^+$ is specific to *Spn*

To determine whether interference with the host NAD$^+$ metabolism is a general process affecting bacterial infections, we performed BEAS-2B infections with *Streptococcus agalacticae* (*S.aga*) and non-typeable *Haemophilus influenzae* (NTHi) and determined the resulting expression of NAMPT and NMNAT1 (Supplementary Fig. 4). Both bacteria induced a significant upregulation of NAMPT expression 16 h post-infection (Supplementary Fig. 4A). *S.aga* infection induced a significant downregulation of NMNAT1 by approximately a factor of 0.75, while NTHi infection, by tendency, caused an upregulation of NMNAT1 (Supplementary Fig. 4B). Next, we investigated the effect of NAD$^+$ treatment on *S.aga*, NTHi and an alternative, highly pathogenic strain of *Spn* (TIGR4). Treatment of *Spn* TIGR4 with NAD$^+$ for 9 h caused a reduction in bacterial growth to approximately factor 0.4, confirming our previous results with *Spn* D39. In contrast, the replication of *S.aga* and NTHi was increased by approximately 50% (Supplementary Fig. 4D–F). In summary, these results reveal that, while the upregulation of NAMPT is a general pro-inflammatory process, the downregulation of NMNAT1 expression and antibacterial effects of NAD$^+$ appear specific to certain bacteria.

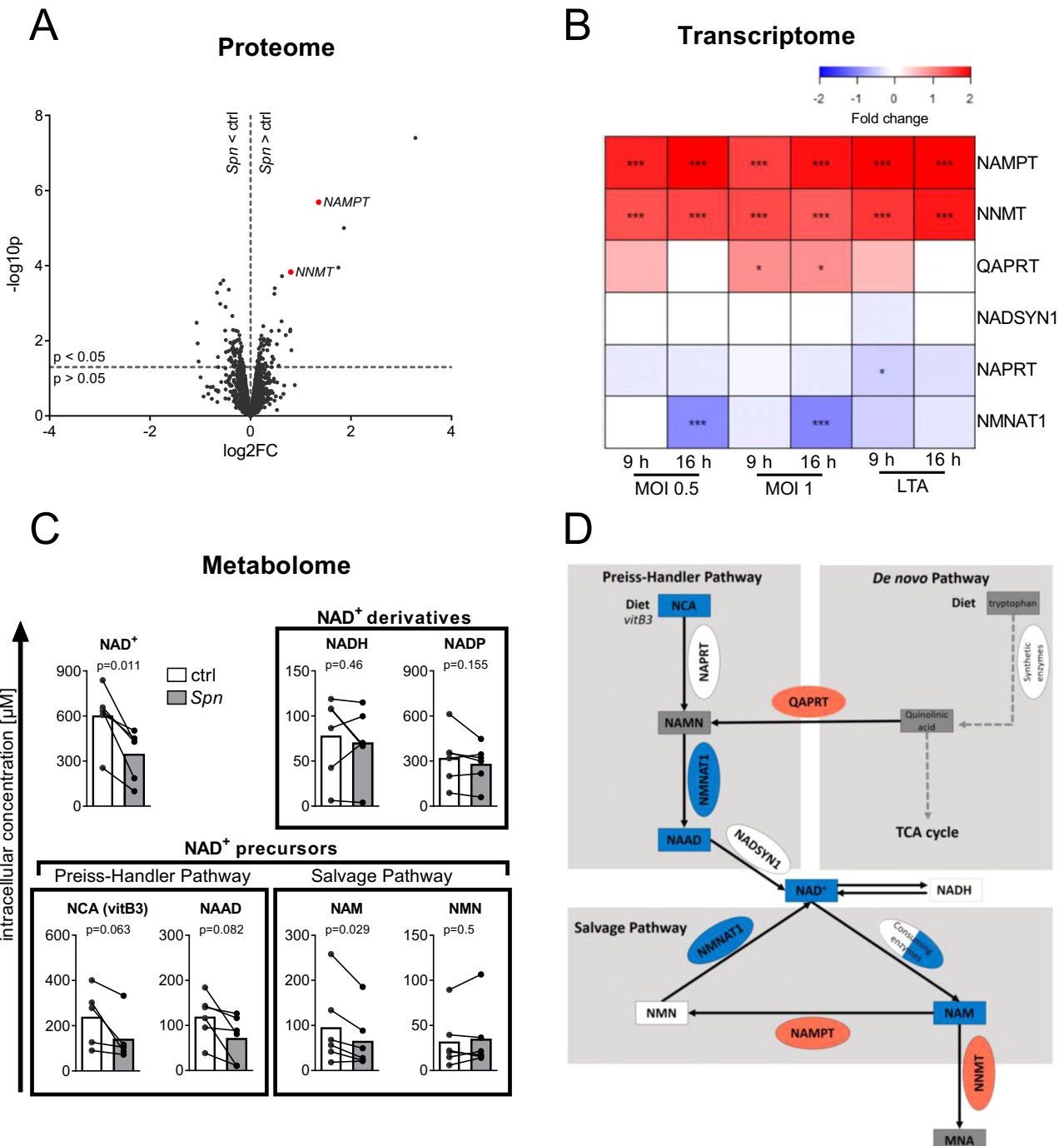

**Fig. 1 | NAD⁺ biosynthesis is dysregulated during *Spn* D39 infection. A** Volcano plot of protein expression upon *Spn* D39 infection. BEAS-2B cells were infected with *Spn* D39, MOI 0.5 for 16 h or left uninfected (ctrl). Afterwards, proteins were isolated and changes in protein expression were determined by SILAC proteomics. Regulated proteins associated with the NAD⁺ metabolism are marked in red. **B** BEAS-2B were either left untreated, infected with *Spn* D39 MOI 0.5 or 1, or treated with 1 μg/ml LTA for 9 or 16 h, respectively. RNA was isolated and gene expression was determined by mRNA microarray. NAD⁺ metabolism-associated genes are depicted as a heat map (value = fold change). **C** BEAS-2B were infected with *Spn* D39 MOI 1 for 16 h or left untreated. Metabolites were isolated and analyzed by LC-MS/MS. **D** Summary of identified regulations of the NAD+ metabolism (red: upregulation; blue: downregulation; square: metabolite; oval: enzyme). Statistics: (**A**, **B**) two-tailed, moderated *t*-test; (**C**) two-tailed paired *t*-test (*$p < 0.05$; ***$p < 0.001$; $N = 3$ biologically distinct samples (**A**, **B**); 6 biologically distinct samples (**C**)). MOI multiplicity of infection.

## Transcriptional regulation of NMNAT1 is induced by *Spn* virulence factor pneumolysin

We next focused on the specific mechanisms by which *Spn* mediates NAD⁺ salvage gene regulation in epithelial cells. Upregulation of NAMPT was previously shown to be induced via JAK-STAT signalling under pro-inflammatory conditions[16]. While we confirmed JAK-dependency of NAMPT upregulation in response to *Spn* D39 infection using the JAK inhibitor Ruxolitinib (Supplementary Fig. 5A), repression of NMNAT1 was not JAK-dependent (Supplementary Fig. 5B). Treatment of epithelial cells with heat-killed *Spn* D39 or LTA isolated from *Spn* D39 did not influence the expression of NMNAT1, suggesting that the repression is mediated by actively produced virulence factors of *Spn* D39

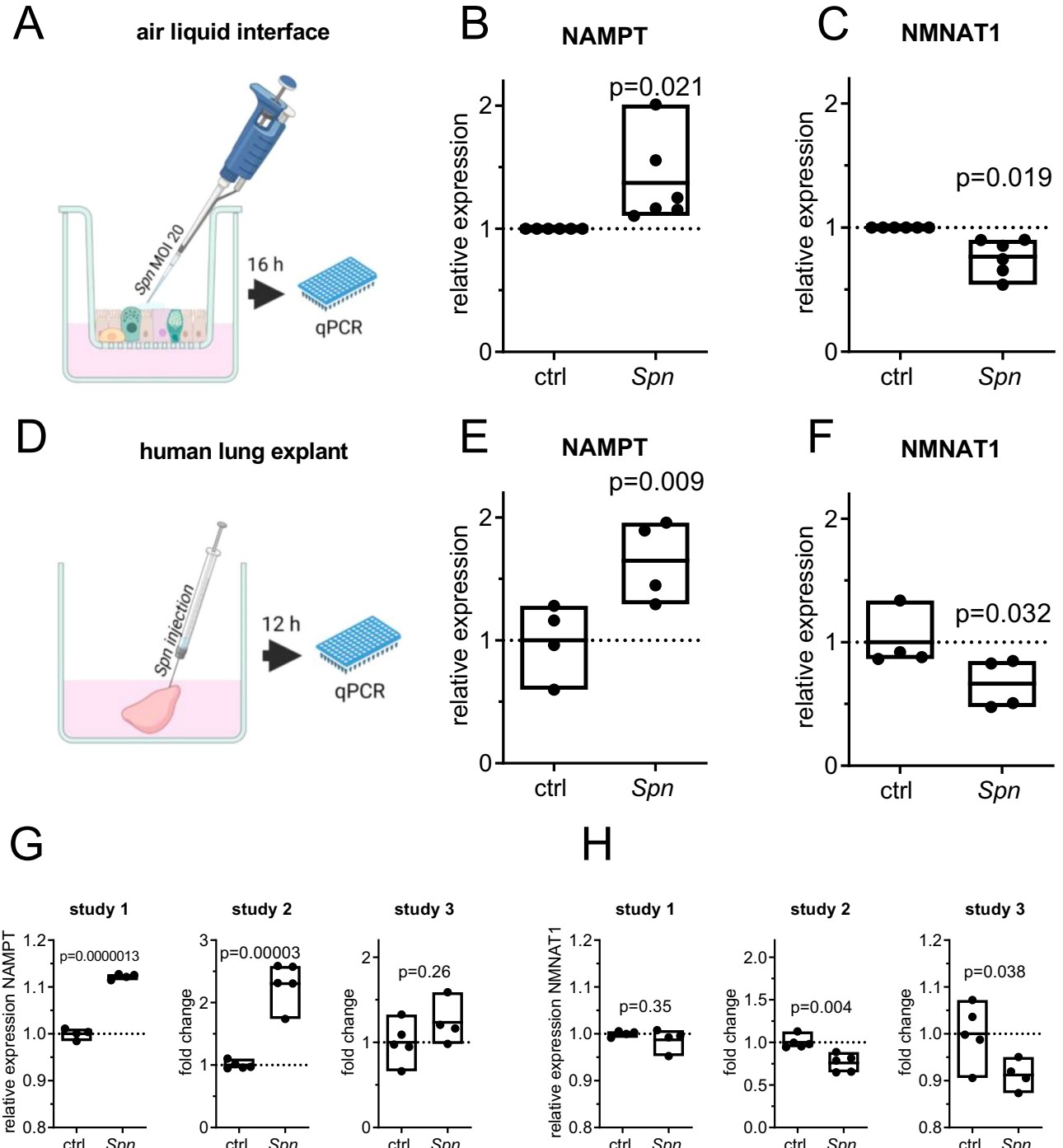

**Fig. 2 | Regulation of NAD⁺ salvage associated genes ex and in vivo. A–C** Primary human bronchial epithelial cells from healthy donors were differentiated into pseudostratified respiratory epithelium as air-liquid interface cultures. After differentiation, cells were apically infected with *Spn* D39, MOI 20 for 16 h. The expression of NAMPT (**B**) and NMNAT1 (**C**) was determined by qPCR. **D–F** Human lung tissue explants were injected with *Spn* D39 or PBS and incubated for 12 h (**D**). Expression of NAMPT (**E**) and NMNAT1 (**F**) was determined by qPCR. **G, H** 3 published microarray transcriptome datasets of murine lungs infected with different strains of *Spn* D39 were analyzed for the expression of NAMPT (**G**) and NMNAT1 (**H**). Study 1: GSE83612, Mouse strain: C57BL/6, male and female; 24 h post-inoculation with $5 \times 10^6$ CFU, serotype: 19; Study 2: GSE45644, mouse strain: BALB/C, female; serotype 2, 48 h post-inoculation with $5 \times 10^4$ CFU; Study 3: GSE61459, mouse strain: BALB/C, female; serotype: 2, 24 h post-inoculation with $5 \times 10^6$ CFU. Statistics: two-tailed paired *t*-test (**B, C, E, F**); two-tailed unpaired *t*-test (**G, H**); $n = 6$ biologically distinct samples (**B, C**); 4 biologically distinct samples (**E, F**) 8 distinct animals (study 1), 9 distinct animals (study 3), 10 distinct animals (study 2) (**G, H**); box plots: line at mean, box ranges from min to max; results are normalized to untreated controls of the corresponding donor (**C, D**) or average expression values of control tissue/animals (**E–H**). Drawings were designed using GraphPad prism. MOI multiplicity of infection.

(Supplementary Fig. 5E–H). Therefore, we investigated the effects of three virulence factors that are actively produced and/or released during pneumococcal growth, the pneumococcal capsule, the oxidant $H_2O_2$ and the pore-forming toxin pneumolysin (*ply*) on NMNAT1 expression. Deletion of the capsule locus (Δ*cps*) and the $H_2O_2$ producing pyruvate peroxidase (Δ*spxB*) did not influence NMNAT1 or NAMPT expression during infection (Supplementary Fig. 5C, D). Infection with a *ply* deficient mutant of *Spn* D39 induced NAMPT gene expression

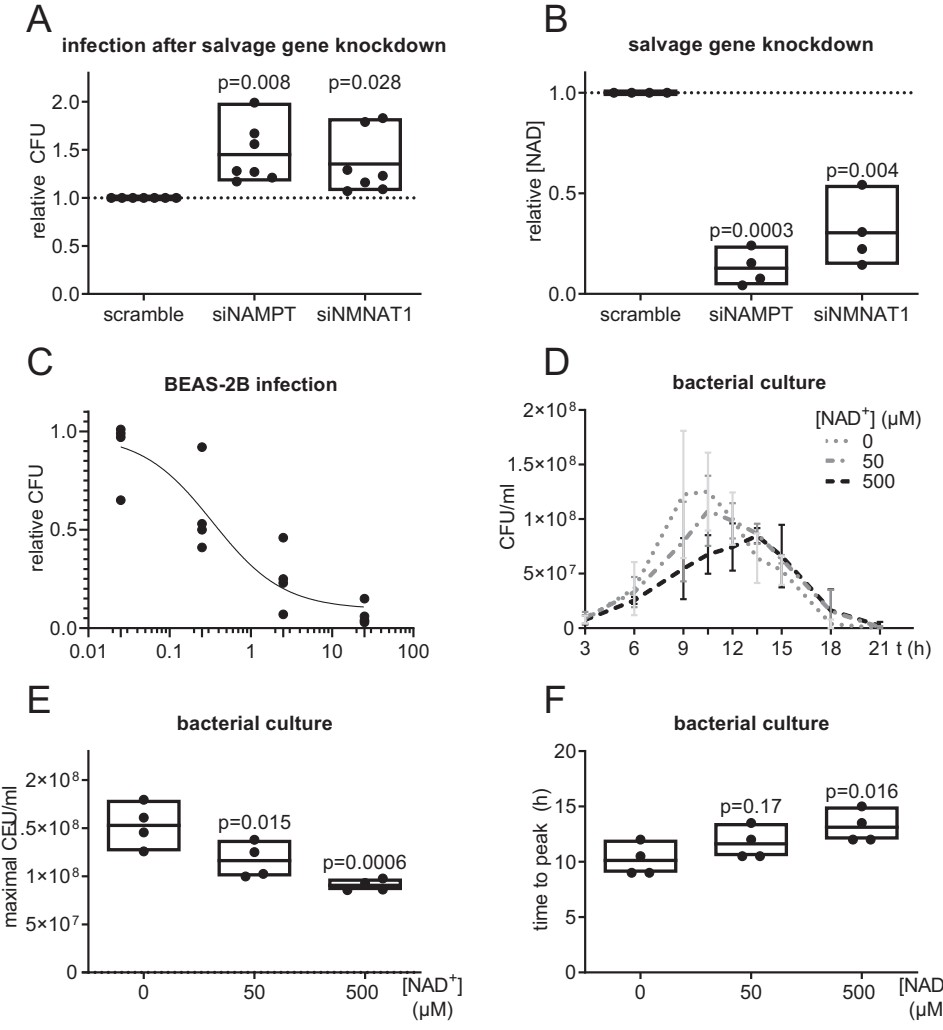

**Fig. 3 | Effects of Nicotinamide metabolites on pneumococcal replication.**
**A** BEAS-2B lung epithelial cells were transfected with indicated siRNA for 48 h prior to infection with *Spn* D39, MOI 1 for 16 h. Bacterial replication was determined by CFU assay. **B** BEAS-2B cells were transfected as indicated for 48 h and the intracellular NAD content determined. **C** BEAS-2B cells were infected with *Spn* D39, MOI 1 for 16 h. Indicated amounts of NAD+ were added at the start of the infection. Bacterial replication was determined by CFU assays 16 h post-infection. **D–F** *Spn* D39 was cultivated in a host cell-free medium with indicated amounts of NAD+. CFU were determined at indicated time points. **D** Bacterial growth curves were measured over 21 h. The graph summarizes the mean of four biological independent replicates. **E** Maximally reached CFU/ml of (**D**). **F** Time (in h) until maximal CFU/ml was reached. Statistics: One-way ANOVA with Fisher's LSD (**A, B, E, F**); Agonist vs Response-Fit (three parameters) (**C**); Error bars indicate SD (**D**); Significances were determined against scramble controls (**A, B**) or untreated controls (**E, F**); $N = 7$ biologically distinct samples (**A**); 4 biologically distinct samples (**B–F**); box plots: line at mean, box ranges from min to max; results are normalized against untreated controls (**A, B**).

to a similar extent as wild-type *Spn* D39 (Fig. 5B). In contrast, a *ply* deletion resulted by tendency in a reduced expression of pro-inflammatory IL-8 (Fig. 5A) and in a significantly attenuated NMNAT1 repression compared to the wild type (Fig. 5C). Treatment of BEAS-2B with purified, lytical active pneumolysin at non-lytic concentrations caused a dose-dependent pro-inflammatory response as indicated by the upregulation of IL-8 and downregulation of both, NAMPT and NMNAT1 (Fig. 5D–F). Interestingly, treatment with a lytical inactive, non-pore-forming variant of pneumolysin did not cause a reduction of NAMPT and NMNAT1 expression (Fig. 5G–I). In summary, pneumolysin might mediate the reduction of NMNAT1 expression via pore formation.

### Resistance against NAD+ is associated with a loss of capsule
We next investigated the antibacterial mechanisms of NAD+. First, we confirmed that extrinsic addition of NAD+ to *Spn* D39 cultures results in increased total intrabacterial NAD (Fig. 6A). Then, resistant clones were generated by cultivating *Spn* D39 in liquid medium with increasing concentrations (50 μM to 5 mM) of NAD+. After six passages, bacteria were plated on blood agar supplemented with 500 μM NAD+ and three clones were picked. All three clones were almost completely resistant against antibacterial effects of NAD+ (Fig. 6B). Genome sequencing revealed the capsule biosynthesis-associated gene *cps2E* to be mutated in all resistant isolates, but not in the control strain (Fig. 6C). The mutation caused an early stop codon in two of three isolates (Clone 1, 3, Fig. 6D). Transmission electron microscopy revealed a complete absence of a capsule for the nonsense-mutated clones 1 and 3.

### NAD+ acts antibacterial by interfering with the pneumococcal energy metabolism
To assess if capsule synthesis and NAD+ sensitivity are causally linked, we performed NAD+ treatment of a capsule-deficient mutant of *Spn* D39. Treatment of *Spn* D39Δ*cps* with NAD+ for 9 h did not show any growth-limiting effects (Fig. 7A). Biosynthesis of the pneumococcal capsule is an energetically costly process and closely linked to

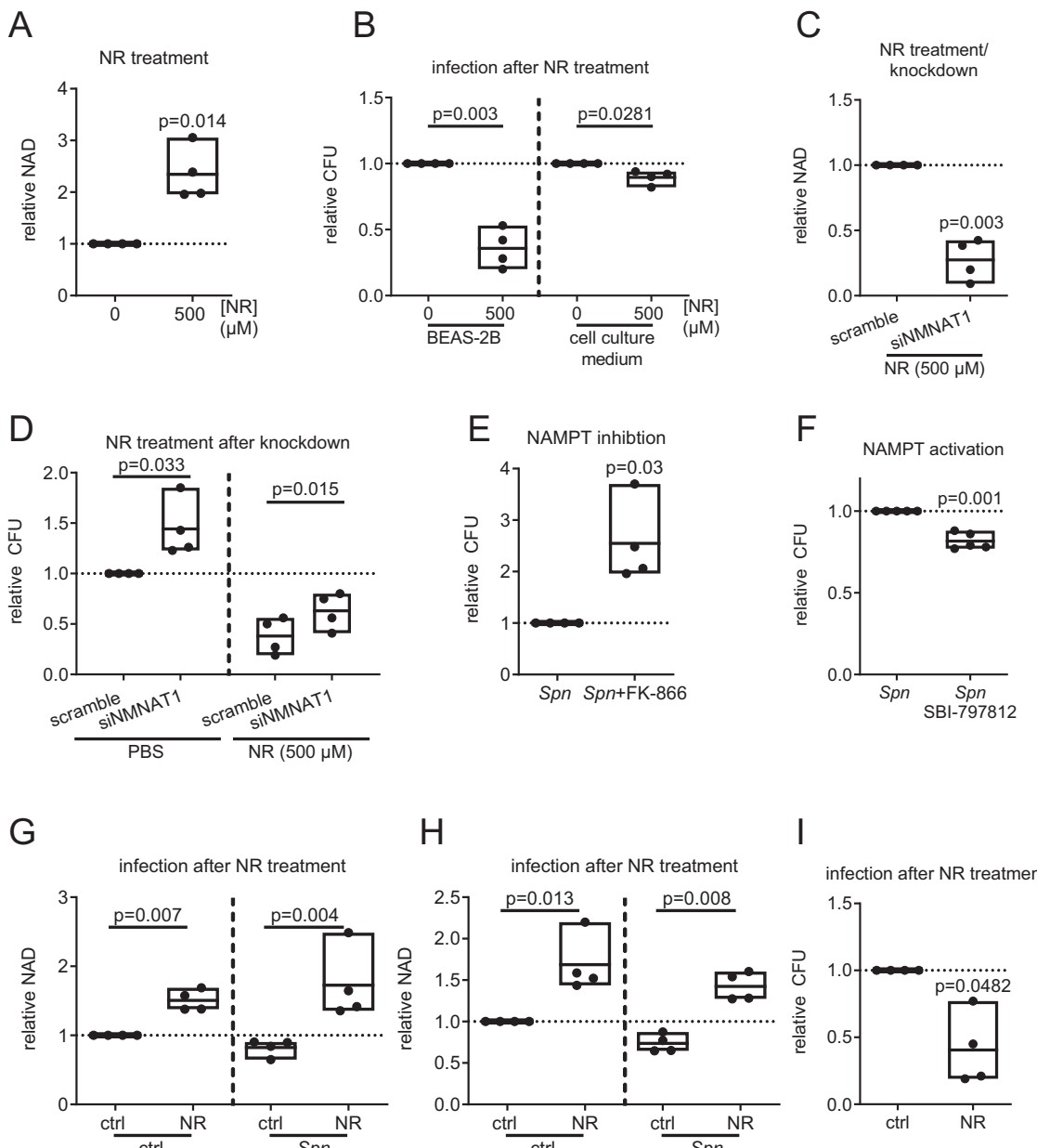

**Fig. 4 | NR-mediated growth inhibition is dependent on NAD⁺ production.**
**A** BEAS-2B were treated with NR for 9 h as indicated and intracellular NAD was measured. **B** *Spn* D39 were cultured in a cell-free medium or used to infect BEAS-2B at MOI 1 with and without NR treatment. CFU were determined after 9 h and are depicted relative. **C** BEAS-2B were transfected with siNMNAT1 or a scramble control for 48 h. Afterwards, cells were treated with NR for 9 h and the intracellular concentration of NAD was determined. **D** Cells were transfected as described. 48 h post-transfection, cells were infected with *Spn* D39, MOI 1 or left uninfected and were treated with NR (500 μM) or PBS. CFU/ml was determined 9 h post-infection and normalized to scramble control. **E** Cells were treated with the NAMPT inhibitor FK-866 (1 mM) 24 h prior to and during infection or left untreated. Bacterial CFU were determined 16 h post-infection. **F** Cells were treated with the NAMPT activator SBI-797812 8 h prior to and during infection or left untreated. Cells were infected

with *Spn* D39, MOI 1 for 16 h and CFU were determined. **G**–**I** Primary pseudostratified human bronchial epithelial cells were cultivated at an air-liquid interface. Cells were infected apically with *Spn* D39, MOI 20 for 1 h. Afterwards, cells were washed, NR (500 μM) was added basolateral and cells were incubated for 16 h. **G**, **H** The amount of intracellular and extracellular NAD was determined 16 h post-infection in cell lysates (**G**) and apical wash fluid (**H**), respectively. **I** Cells were washed 16 h post-infection and CFU in the apical wash fluid was determined. Statistics: two-tailed paired *t*-test; significance was determined against uninfected/untreated controls unless indicated otherwise; $N = 4$ biologically distinct samples (**A**–**E**, **G**–**I**); 5 biologically distinct samples (**F**); box plots: line at mean, box ranges from min to max; error bars indicate min. and max. value (**D**); results are normalized against untreated controls. NR nicotinamide riboside, hbec human bronchial epithelial cells.

bacterial energy metabolism[17]. Under energetically detrimental conditions, the capsule was shown to interfere with pneumococcal growth[18]. We therefore hypothesized that NAD⁺ interference with intrabacterial ATP homeostasis is responsible for the growth-limiting effect of NAD⁺. *Spn* D39 WT, Δ*cps*, and NAD⁺-resistant clone 3 all showed a significant reduction in intrabacterial ATP after NAD⁺ treatment (Fig. 7B). When left untreated, *Spn* D39 clone 3 and *Spn*

D39Δ*cps* showed an increase in intrabacterial ATP by a factor of 2 compared to the WT strain (Fig. 7B). Treatment of bacteria with 5 mM of the ATP precursor pyruvate in addition to NAD⁺ treatment abolished the NAD⁺-dependent reduction in intrabacterial ATP and reduction in bacterial CFU (Fig. 7C, D). In summary, the NAD⁺-mediated antibacterial effects were dependent on shifts in bacterial energy metabolism and capsule biosynthesis.

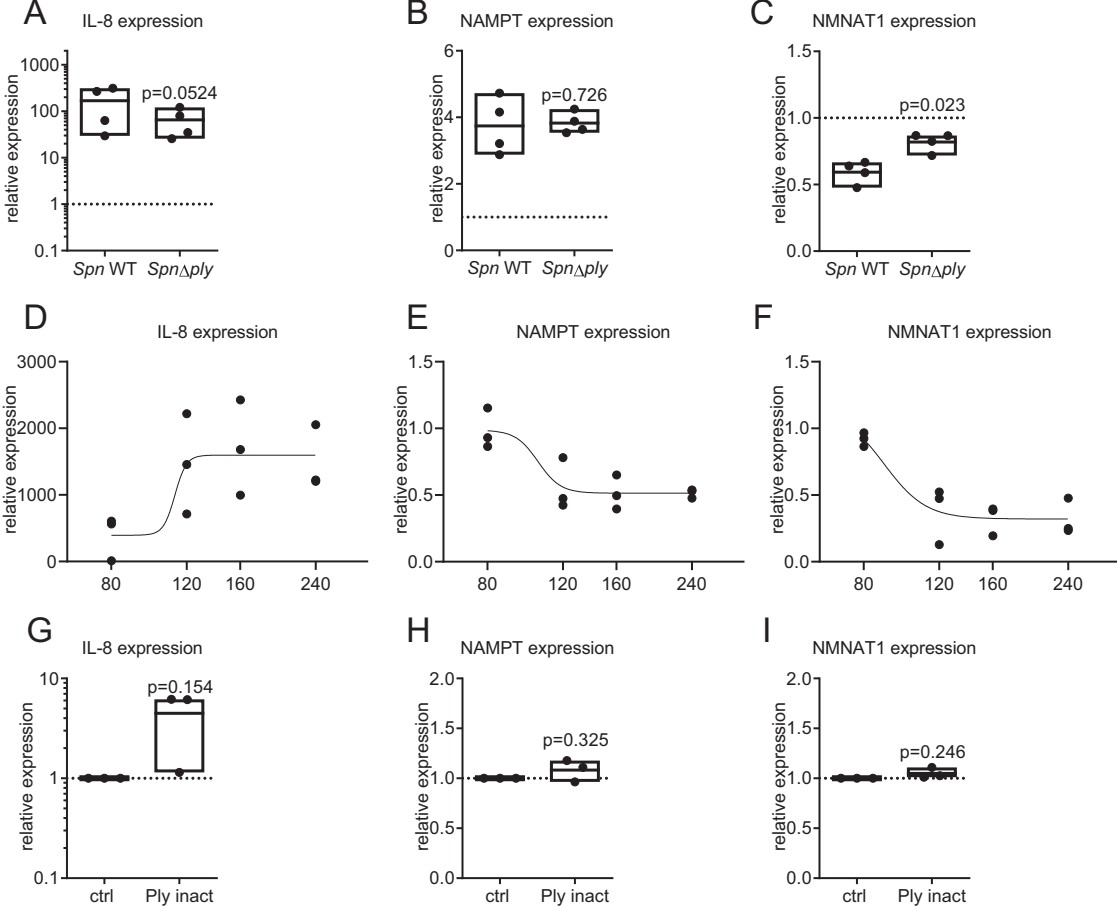

**Fig. 5 | Transcriptional regulation of NMNAT1 is induced by pneumolysin (ply).**
**A–C** BEAS-2B cells were infected with *Spn* D39, WT or Δ*ply* for 16 h. Gene expression was determined by qPCR. **D–F** BEAS-2B cells were treated with pneumolysin in indicated concentrations for 16 h. Expression of IL-8, NAMPT and NMNAT1 was determined as indicated. **G–I** BEAS-2B cells were treated with 120 nM inactive pneumolysin for 16 h. Expression of IL-8, NAMPT and NMNAT1 was determined as indicated. Statistics: two-tailed paired *t*-test (**A–C**, **G–I**); sigmoidal dose-response fit with variable slope (**D–F**); Significances were determined against wild-type infected (**A–C**) or unstimulated controls; $N = 4$ biologically distinct samples (**A–C**) 3 biologically distinct samples (**D–I**); box plots: line at mean, box ranges from min to max; results are normalized against untreated controls. Ply pneumolysin.

## Discussion

In this study, we describe a previously unreported interference of *Streptococcus pneumoniae* with the host NAD metabolism. Our multi-omics approach revealed that in response to *Spn* D39 infection, NAMPT and NNMT are upregulated in bronchial epithelial BEAS-2B cells as part of a general, pro-inflammatory response. This is counteracted by a *Spn*-specific mechanism to downregulate NMNAT1 expression. The observed gene regulations seem to be pathophysiological relevant, as primary human bronchial epithelial cells cultivated at air-liquid interface, human lung explants and lungs of *Spn*-infected mice showed similar gene regulations. Since NMNAT1 catalyzes the final step of the salvage pathway and NNMT transforms NAM into the non-salvageable metabolite MNA, the identified gene regulations result in a reduced intracellular concentration of NAD[+].

Here we report a *Spn*-mediated interference with the host NAD[+] homeostasis. NAD[+] was, however, shown to be a major player during other infections. During the viral infection process, NAMPT is routinely upregulated in a JAK-STAT-dependent manner. This results in an increased NAD[+] biosynthesis to fulfil the energetic demands of the antiviral enzymatic machinery[19]. To evade this defence mechanism, several viruses synthesize macrodomains which counteract the function of NAD[+]-consuming PARP by hydrolyzing ADP ribosylation sites of host proteins[20,21]. In contrast, the murine hepatitis virus, a model coronavirus, was shown to deplete the cellular NAD[+] storage by upregulating PARP activity[22]. Direct interference of viruses with NAMPT

expression was suggested to be mediated by the induction of miRNA expression, which prevents NAMPT translation[23,24]. During bacterial infections, several bacterial pathogens, among them *Shigella flexneri* and the lung pathogens NTHi and *Mycobacterium tuberculosis*, were shown to utilize host-produced NAD[+] for their own energy metabolism[25–27]. In accordance, with this study, NTHi infection induced NAMPT expression which might cause an increase in NAD[+] production. As NTHi does not contain LTA, this effect might be associated with the Outer Membrane Protein 2 (porin), which was previously shown to activate TLR2[28]. Besides that, bacteria of the gut microbiota were shown to produce nicotinic acid as an NAD[+] precursor to enhance host NAD[+] production[29].

In contrast to these previously described upregulations of host NAD[+] production by different bacteria, we here describe a *Spn* specific downregulation of the NAD[+] biosynthesis via repression of NMNAT1 gene expression mediated by pneumolysin. To assess how this affects the infection process, we experimentally decreased NAD[+] biosynthesis by knockdown of NAMPT or NMNAT1 prior to bacterial infections. Knockdown of NAMPT or NMNAT1 both reduced epithelial NAD production and increased bacterial replication. NAMPT is a multifaceted enzyme with previously reported cytokine function during pro-inflammatory processes[30]. NMNAT1, in addition to its biosynthetic function, was reported to act as a chaperone[31]. To our knowledge, however, no involvement of NMNAT1 in infection processes has been reported before. To determine whether NAD[+]

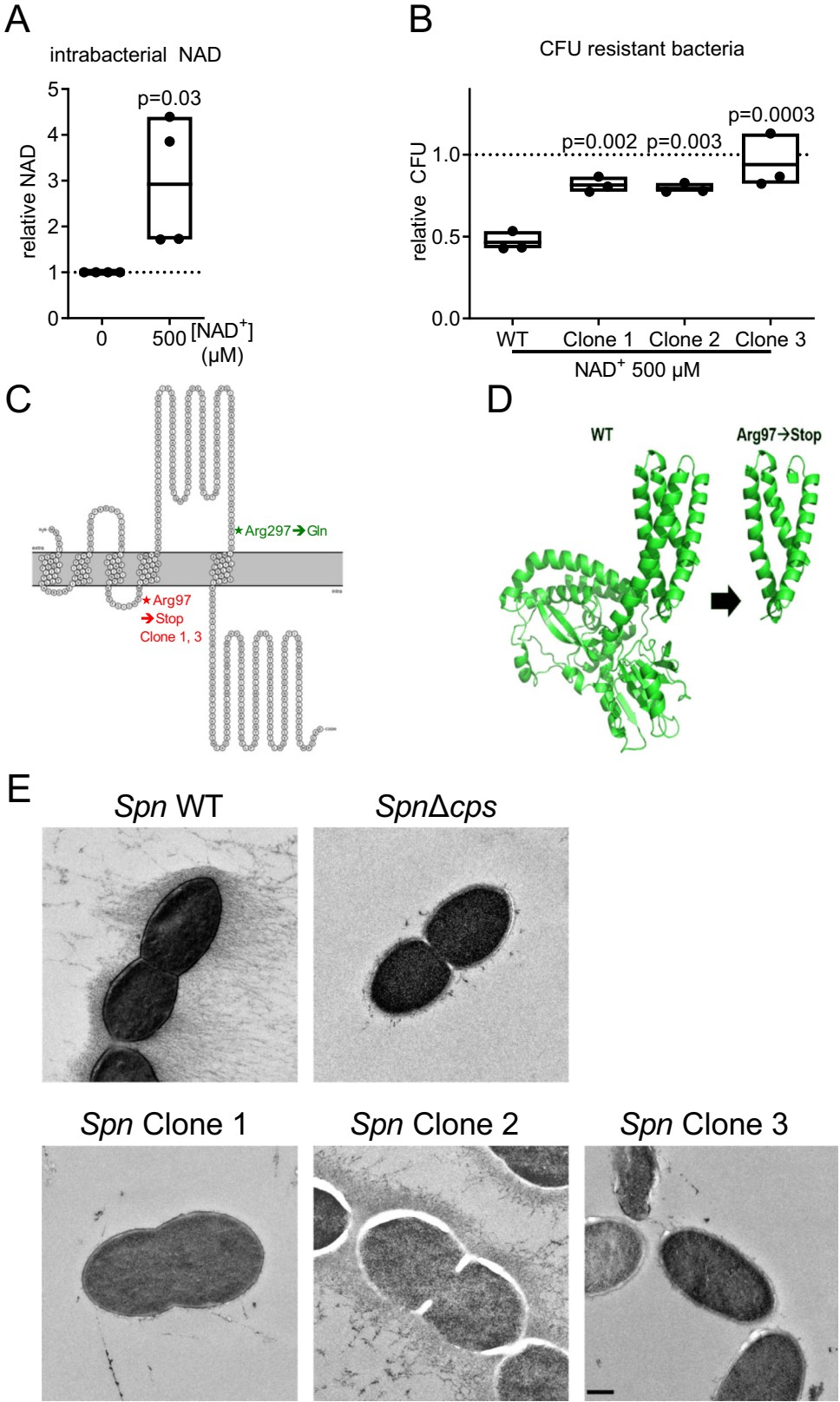

biosynthesis or secondary effects of NAMPT and NMNAT1 are responsible for the observed effect, we treated *Spn* D39 with $NAD^+$ in various concentrations. This resulted in a host-cell independent reduction of bacterial replication[+]. Interestingly, the precursors NAM, NMN and NR showed no or minor inhibition of bacterial replication. Together, these results demonstrate a direct antibacterial effect of $NAD^+$. It was shown that treatment with the $NAD^+$

precursor NR increases the intracellular concentration of $NAD^+$ in vitro and in vivo[13–15]. Boosting cellular $NAD^+$ production by NR and NAM treatment was shown to inhibit murine herpes virus replication in vitro[22] and connected to the energetic demand of the antiviral cellular machinery. Accordingly, replenishing intracellular $NAD^+$ was proposed as an immune-enhancing pharmacological invention in the treatment of viral infections[32]. To confirm that epithelial NMNAT1 can

**Fig. 6 | Development of NAD⁺ resistance is associated with loss of capsule. A** *Spn* D39 was inoculated in a cell culture medium with and without NAD⁺ treatment as indicated. Bacteria were lysed 6 h post-inoculation and the total intracellular NAD was determined. **B–D** *Spn* D39 was cultivated in a cell culture medium and passaged 6 times with increasing concentrations of NAD⁺(50 µM to 5 mM) or without treatment. Afterwards, bacteria were plated on NAD⁺ (500 µM) supplemented agar plates. Individual clones were picked and characterized for NAD⁺ resistance and growth behaviour and sequenced. **B** *Spn* D39 WT and resistant bacteria were treated with 500 µM NAD⁺ for 6 h and bacterial counts were determined. **C** *Spn* D39 clones passaged with increasing concentrations of NAD⁺ and a control passaged in cell culture medium

without NAD⁺ were sequenced. Identified amino acid changes in the protein CPS2E are displayed. Asterisks indicate amino acid substitutions. **D** 3D protein structure of CPS2E wild type and Arg97→Stop, generated with AlphaFold. **E** Transmission electron microscopy of *Spn* D39 WT, Δ*cps* and the NAD⁺ resistant clones. Bacteria were cultivated until the early logarithmic phase in liquid media, cryo-fixated and the capsule stained using $OsO_4$. Statistics: two-tailed paired *t*-test (**A**); One-way ANOVA with Fisher's LSD (**B**); significance was determined against the untreated control (**A**) or the wild-type bacteria (**B**); Scale: 250 nm; *N* = 4 biologically distinct samples (**A**); 3 biologically distinct samples (**B**); box plots: line at mean; box ranges from min to max; results are normalized against untreated controls. WT wild type.

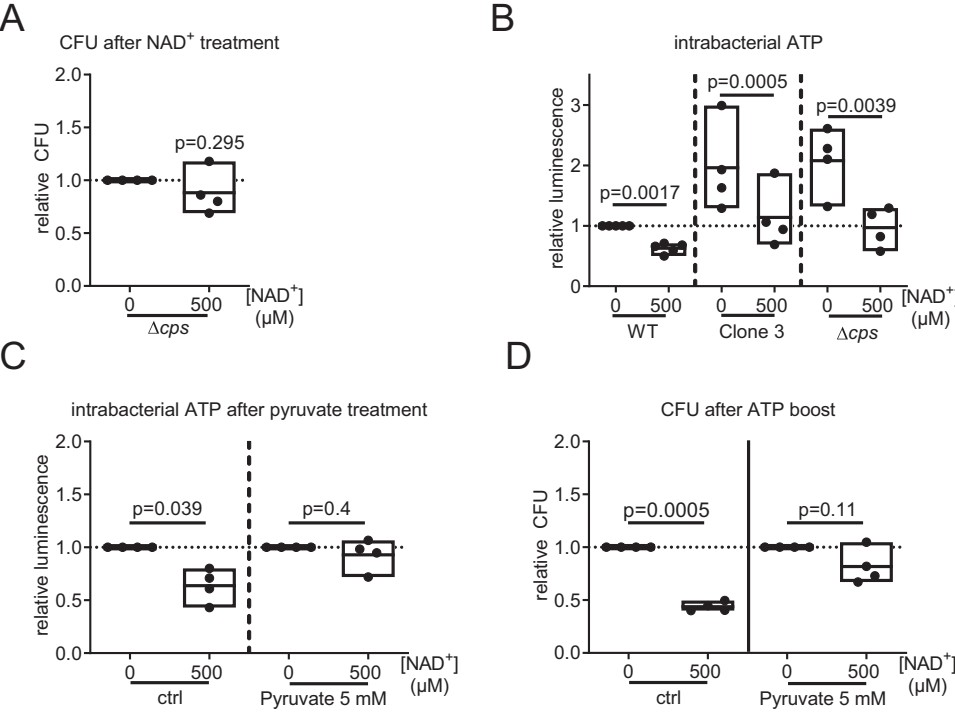

**Fig. 7 | Development of NAD⁺ resistance is associated with metabolic adaptation. A** *Spn* D39Δ*cps* was treated with NAD⁺ for 9 h or left untreated and bacterial counts were determined. Relative bacterial counts are depicted. **B** *Spn* D39 WT, clone 3 and *Spn* D39Δ*cps* were incubated in a cell culture medium for 6 h with and without NAD⁺ treatment. Afterwards, the intrabacterial ATP was determined by a commercial luminescence assay and normalized against bacterial counts. Measured luminescence is linear to intrabacterial ATP. **C, D** Intrabacterial ATP after 6 h

(**C**) and bacterial counts after 9 h (**D**) of combined treatment of *Spn* D39 WT with 500 µM NAD⁺ and 5 mM pyruvate. Statistics: two-tailed *t*-test (**A**, **B**); Two-way-ANOVA with Fisher's LSD (**C**, **D**); significance was determined against uninfected/untreated controls if not indicated otherwise; *N* = 4 biologically distinct samples; box plots: line at mean; box ranges from min to max; results are normalized against untreated controls. WT wild type.

produce sufficient NAD⁺ to exert antibacterial activity, we supplemented host cells with NR during infection resulting in a significant increase of intracellular NAD. Similar results were obtained when boosting host cell NAD⁺ synthesis by treatment with the precursors NMN and NAM. The NR, NMN and NAM-mediated inhibition of bacterial growth was almost or completely abolished in the absence of host cells, suggesting an involvement of host cell NAD⁺ biosynthesis in the antibacterial effects of NR. Knockdown of NMNAT1 in BEAS-2B prior to infection and NR treatment suppressed the cells' ability to utilize NR for NAD⁺ synthesis. Following NMNAT1 knockdown, intra-host cell concentration of NAD⁺ was reduced and bacterial replication was partially rescued, confirming the role of NMNAT1 in antibacterial NAD⁺ production. The remaining growth-inhibitory effects of NR might be due to the observed minor antibacterial effect of NR, remaining NMNAT1, or enzymatic activity of the related enzymes NMNAT2 and NMNAT3[33,34]. Results were confirmed when treating BEAS-2B cells with NMN and NAM, both of which did not exhibit any antibacterial activity in the absence of host cells. Additionally, NMN treatment of primary human bronchial epithelial cells cultivated at an air-liquid interface equally increased intracellular NAD and reduced

bacterial replication, supporting the relevance of the identified mechanism in physiological settings. Likewise, treatment of BEAS-2B cells with the NAMPT inhibitor FK-866[35] and the NAMPT activator SBI-797812[36] resulted in increased and decreased *Spn* D39 replication, respectively, without requiring an extrinsic NR boost. This demonstrates the intrinsic potential of the NAD⁺ salvage pathway to maintain antibacterial NAD⁺ concentrations.

While a previous study showed shifts in metabolic gene expression of *Spn* D39 upon NADH treatment, no growth-limiting effects were observed[37]. Besides that, the effect observed in this study is not limited to the *Spn* D39, as treatment of *Spn* TIGR4 with NAD⁺ equally inhibited bacterial growth. It is, however, not a general antibacterial mechanism, as *S.aga* and the host ATP-dependent NTHi[38] showed improved growth upon NAD⁺ treatment. *Spn* D39 frequently appears as an asymptomatic colonizer of the upper airways. The bacteria-specific responses to dysregulated NAD⁺ production raises the interesting question of whether and how NAD⁺ is involved in the interaction between *Spn*, airway microbiota and the host. In summary, boosting the host-cell NAD⁺ metabolism specifically inhibits pneumococcal replication during the infection process.

Since NAMPT was previously reported to be regulated via the JAK-STAT pathway[16,39], we hypothesized NMNAT1 to be regulated via a similar pathway. However, while in accordance with the literature inhibition of JAK signaling by Ruxolitinib[40] reduced NAMPT expression after *Spn* D39 infection, expression of NMNAT1 was not affected. In the following, we assessed which pneumococcal virulence factor is responsible for the observed gene regulation. Virtually all pathogenic strains of *Spn* express the pore-forming cholesterol-dependent cytolysin pneumolysin[41,42]. Besides its cytolytic effect, pneumolysin was shown to induce TLR4-dependent and -independent pro-inflammatory responses and cause DNA damage in the host cell[43–45]. Deletion of pneumolysin from *Spn* D39 rescued NMNAT1 expression in BEAS-2B after infection, while NAMPT expression remained unaffected. Furthermore, treatment of BEAS-2B with purified pneumolysin in sublytic concentrations caused a dose-dependent downregulation of NMNAT1 and NAMPT. The effects were dependent on lytic activity, as a lytical inactive variant of pneumolysin did not affect NMNAT1 and NAMPT expression. Interestingly, *S.aga*, which produces another pore-forming toxin, caused the downregulation of NMNAT1 similar to *Spn*[46]. These results support the hypothesis of dysregulation of host NMNAT1 expression as a pneumococcal mechanism to inhibit detrimental NAD$^+$ production in the host. In contrast, the observed downregulation of NAMPT during treatment with purified pneumolysin might be masked by the pro-inflammatory JAK-STAT-dependent upregulation of NAMPT. Intracellular bacteria like *Mycobacterium tuberculosis* were previously reported to export toxins into the cytoplasm which deplete host-cell NAD$^+$ and interfere with the host metabolism[47,48]. Related bacteria like Group A *Streptococcus* were shown to express NAD$^+$ glycohydrolases which deplete intra-host-cell NAD$^+$ and might thereby defend themselves against NAD$^+$ mediated bacterial killing[49]. For a pore-forming cytolysin, to our knowledge, interference with the host cell energy metabolism has not been reported before.

To gain insight into the antibacterial mechanism of NAD$^+$, we generated three NAD$^+$ resistant strains of *Spn* D39 and compared their genome sequence to the wild type. All sequenced isolates showed mutations in the gene encoding for the protein CPS2E. CPS2E catalyzes an initial step in the attachment of the pneumococcal capsule to the membrane. Amino acid exchanges in this protein were previously shown to render *Spn* D39 unencapsulated[50]. The lack of capsule for two of the generated strains, both containing an early stop codon, was confirmed by electron microscopy. In line with this observation, a *Spn* D39 strain with deleted *cps* locus was resistant against NAD$^+$ treatment in our experimental setting. The pneumococcal polysaccharide capsule surrounds the whole bacterial cell and is a major virulence factor, protecting *Spn* against adverse environmental conditions and the immune defense[51]. Capsule production, however, is a major biosynthetic effort and closely intertwined with the bacterial energy metabolism[17,52]. Starvation of *Spn* leads to a reduction in capsule size[53] and under Cadmium-induced metabolic stress *Spn* was shown to reduce capsule production[53,54]. We therefore hypothesized that abolishing capsule production is an adaptation to a dysregulated energy metabolism. To test our hypothesis, we measured intrabacterial ATP with and without NAD$^+$ treatment. NAD$^+$ treatment of *Spn* D39 WT resulted in a reduction of intrabacterial ATP. Strikingly, an NAD$^+$ resistant strain and *Spn* D39Δ*cps* showed an approximately two-fold increased amount of basal intrabacterial ATP, supporting the link between NAD$^+$ resistance and ATP homeostasis. We then aimed to abolish the dysregulation of ATP production by treating bacteria with the ATP precursor pyruvate in parallel to NAD$^+$ treatment. Indeed, pyruvate treatment increased intrabacterial ATP and abolished the antibacterial effects of NAD$^+$. It is important to note that NAD$^+$ cannot freely diffuse through membranes. Bacteria like *H. influenzae* were previously shown to degrade NAD$^+$ and take up the degradation products. In this study, however, the NAD$^+$ precursor NR, NAM and NMN did not exhibit major antibacterial activity, suggesting a direct antibacterial mechanism of NAD$^+$. Different

nucleotide uptake mechanisms were reported in bacteria and yeast. Nucleotide exchange transporters were described to be involved in NAD$^+$ uptake in chlamydia[55] and yeast[56]. In peroxisomes, an NAD/AMP exchange transporter was reported to import NAD$^+$ across membranes[57] and a similar process might be involved in NAD$^+$ uptake and metabolic dysregulation in *Spn*. Furthermore, *Spn* is well known for its natural competence, its ability to take up DNA fragments via membrane channels and ATP-dependent translocases. This process might be involved in the uptake of NAD$^+$ and reduction of intrabacterial ATP levels[58,59]. Alternatively, NAD$^+$ interference with the bacterial ATP homeostasis might be associated with the recognition of NAD$^+$ by bacterial surface receptors or interference with ATP-consuming processes on the bacterial cell surface[60]. Interestingly, in contrast to other NAD$^+$ precursors, NR caused a mild reduction of bacterial replication in the absence of cells. NR can likely be imported into *Spn* by PnuC[61]. While the growth-limiting effect of NR was minor compared to direct NAD$^+$ treatment of bacteria or the growth limitation in the presence of cells, uptake of NR as a degradation product of NAD$^+$ might further be involved in NAD$^+$ mediated growth-limiting effects against *Spn*. Collectively, these data highlight the link between NAD$^+$ stress, capsule biosynthesis, and bacterial metabolic dysregulation, and provide a mechanism for the antibacterial activity of NAD$^+$.

There are limitations to our study. It is difficult to determine how the NAD$^+$ concentrations used in this study relate to the actual extracellular concentration of NAD$^+$ in vivo. In human blood, NAD$^+$ concentrations were determined to be approximately 33 μM[62]. While this is below the threshold for significant growth inhibition of *Spn* in our study, concentrations might be higher in the infection-relevant microenvironments like the lung alveoli, especially upon infection-induced host cell lysis[63].

Furthermore, extensive animal experiments to confirm increased NAD$^+$ concentrations after precursor supplementation and its effect on bacterial replication were beyond the scope of this study. Nevertheless, we were able to confirm the NAD$^+$ salvage dysregulation in infected mouse lungs and showed the antibacterial effects of NAD$^+$ production in ex vivo human lung infection models. Since an increase of NAD$^+$ levels following precursor treatment in vivo is well established[12,64], it seems likely that the antibacterial effect of increased NAD$^+$ production by nicotinamide precursor supplementation will hold up in vivo.

While, this study investigated the effect of differentially regulated host NAD$^+$ biosynthesis on the infecting bacteria, changes in metabolite homeostasis and repressed NAD$^+$ production are likely to have important consequences on host cell function as well. Interestingly, in addition to NAMPT and NMNAT1, the expression of NNMT was upregulated in BEAS-2B. MNA, the product of the enzymatic reaction catalyzed by NNMT was previously shown to exhibit anti-inflammatory effects in mice and humans[65,66]. Elevated amounts of MNA were furthermore shown to increase THP-1 proliferation and are associated with negative outcomes in cancer patients. These results thereby further highlight the importance of the nicotinamide metabolism during pneumococcal infections.

Finally, NAD$^+$ is an important energy source for antiviral host responses[19] and is expected to be similarly important for mounting antibacterial responses. Reduced intracellular NAD$^+$ and therefore reduced activity of NAD$^+$-dependent sirtuins was associated with aging processes[67]. It was recently proposed that residential bacteria contribute to Alzheimer's disease and progression[68]. Should the observed dysregulation of NAD$^+$ production continue after bacterial clearance, this raises the fascinating question of whether pneumococcal infections, especially pneumonia and meningitis, cause long-term detrimental metabolic effects, contributing to accelerated lung or brain ageing. The same question holds for other viral and bacterial infections that interfere with the expression of genes involved in NAD$^+$ biosynthesis.

Taken together, our study reveals NAD$^+$ biosynthesis as a previously overlooked antibacterial defence mechanism against *Streptococcus*

*pneumoniae* and sheds further light on the importance of the energy metabolism as a key playing field of host–pathogen interaction during infection.

## Methods

### Ethics statement

Bronchial tissue for cell isolation was kindly provided by the Biobank platform of the German Center for Lung Research (DZL, Biobank Giessen, Hesse, Germany). Donated tissue was handled in accordance to local ethics regulations (Philipps Universität Marburg; permit number: AZ 224/12 for work with primary human bronchial epithelial cells; permit number AZ 161/17 for work with lung explants) and analysed anonymously. Informed consent was obtained from all donors.

### Statistics and reproducibility statement

No statistical calculation was performed to determine sample size. Instead, the number of replicates was chosen based on previously published studies[69,70], our experience with the infection experiments and the observed effect strength. In general, at least 3 replicates were performed. At least 4 replicates using material from 4 different donors were performed when primary epithelial tissue was used for experiments. Additional N were performed in case of small effect sizes or high variation between replicates. 6 N were performed for metabolome analysis, as the metabolites in question can be highly volatile[71]. Replicates were excluded if they did not match quality control criteria (RNA quality, IL-8 expression). When immortalized cell lines were used, experiments were reproduced with at least 3 different passages of cells. Observed gene regulations were further reproduced using two different batches BEAS-2B cells obtained by ATCC and Merck, respectively. Investigator blinding was not performed in general, but investigators were blinded during analysis of metabolite MS results and electron microscopy.

Statistical analysis of results was performed using GraphPad Prism 9.5 (GraphPad Software) and R 4.3 with the limma package. Statistical tests and number of replicates are indicated in the figure legends.

### Bacterial strains and culture

A clinical isolate of *S.aga* obtained at the University Medical Center Marburg was kindly provided by Frank Sommer (Phillips University Marburg, Germany), whereas NTHi 19418 was obtained from ATCC. *Spn* and *S.aga* were plated on Columbia Blood agar plates (Becton Dickinson, 254005) and incubated over night at 37 °C/5% $CO_2$. Single colonies were inoculated into Todd-Hewitt-Yeast (THY) media and incubated at 37 °C/5% $CO_2$ until early logarithmic phase ($OD_{600}$ 0.3–0.4). Bacteria were sedimented (3000 × *g*, 15 min, RT) and adjusted to OD 0.1 in BEGM, corresponding to $5 × 10^7$ CFU per ml (*Spn* D39) or $2 × 10^7$ CFU/ml (*S.aga*), respectively. NTHi was plated on Chocolate Blood agar plates (Becton Dickinson, 257011) and incubated over night at 37 °C/5% $CO_2$. Single colonies were inoculated into Brain-Heart infusion (BHI) media, supplemented with Hemin and NAD+ (Merck, N7004) and incubated at 37 °C/5% $CO_2$ until early logarithmic phase ($OD_{600}$ = 0.3–0.4). Bacteria were sedimented (3000 × *g*, 15 min, RT) and adjusted to OD 0.1 in BEGM; corresponding to $5 × 10^7$ colony forming units (CFU)/ml. *Spn* D39 Δ*cps*[72], Δ*ply*[73] and Δ*spxB*[74] were described in previous work. To generate NAD+ resistant strains of *Spn* D39, bacteria were cultivated in THY medium. After 3 h of incubation per step, they were continuously diluted to fresh medium with gradually increasing concentrations of NAD+ (50 μM, 100 μM, 250 μM, 500 μM, 1 mM).

### Bacterial infection and cell stimulation

For BEAS-2B infection experiments, bacteria were inoculated into fresh BEGM (Lonza, CC-3170) medium according to desired multiplicity of infection (MOI). Alternatively, cells were treated with the putative TLR2 ligand lipoteichoic acid from *Staphylococcus aureus* (LTA, 1 μg/ ml, Invivogen, tlrl-pslta), the pore-forming cytolysin pneumolysin (Ply), Ruxolitinib (10 μM, SelleckChem, S1378), FK-866 (1 μM, R&D Systems, 4808) or SBI-797812 (1 μM, Merck, SML2791) whereas control cells were treated with the respective dissolvent.

### CFU assay and bacterial growth curve

For analysis of bacterial replication during infection, supernatant of infected cells was collected at indicated time points and 10-fold serial dilutions were streaked on blood agar plates. After incubation overnight at 37 °C and 5% $CO_2$, bacterial colonies were counted manually and CFU/mL were calculated.

To assess direct inhibitory effects of metabolites on *Spn*, $5 × 10^5$ CFU of *Spn* in early logarithmic growth phase were inoculated in BEGM and treated with NAD+, NR (Sigma-Aldrich, SMB00907), NAM (Sigma-Aldrich, 72340) or NMN (Sigma-Aldrich, N3501). Bacterial cultures were cultivated at 37 °C/RT for indicated times and the CFU assay was performed as described above.

### Bacterial sequencing

Total DNA for sequencing was purified using a DNeasy Blood and Tissue kit (Qiagen). DNA libraries for sequencing were generated by applying a Nextera XT DNA Library Preparation kit (Illumina, FC-131-1024), and sequencing was performed on a MiSeq Desktop Sequencer (Illumina) using a MiSeq Reagent kit, version 3, for 2 75-bp paired-end reads (Illumina, MS-102-3001). At least 5.3 mio reads were obtained for each sample. The investigation for single nucleotide variants was carried out using the Basic Variant Detection tool (Qiagen, v.2.2) of CLC Genomic Workbench (Qiagen, v.21.0.3) with a minimum coverage of 10, minimum count of 8 and minimum frequency of 80% for mapped reads.

### Electron microscopy

For electron microscopy, *Spn* D39 strains were inoculated into THY medium and incubated at 37 °C. When $OD_{600}$ reached 0.35, the bacterial culture was centrifuged at 8000 g for 10 min. A small pellet of bacteria was cryo-fixed between two golden 75 μm apertures in liquid ethane using the sandwich plunge freezing method[75] and freeze-substituted in 2% osmium tetroxide (Sigma, 201030), 0.1% uranyl acetate, and 5% distilled water in acetone using the fast low-temperature dehydration and fixation method. Bacteria were infiltrated overnight in Epon resin (LADD, NC9925769) and polymerized at 60 °C for 48 h. 80-nm-thick sections were cut with a Leica UC6 ultramicrotome and imaged with a Tecnai T12 (FEI, Eindhoven) transmission electron microscope running at 120 kV.

### ATP measurement

Total concentration of intra-bacterial ATP was determined using a luminescence assay (Promega, G8230) according to manufacturer's instructions. ATP concentrations were normalized to bacterial counts as determined by CFU assay.

### Purification of Pneumolysin

LPS-free pneumolysin was overexpressed in a recombinant *Listeria innocua* 6a strain. Lytical inactive pneumolysin was generated by deleting alanine at position 146[76]. Proteins were expressed with an N-terminal histidine tag and purified by affinity chromatography. Purity was assessed by Western Blot and MS analysis. The batch of wild type pneumolysin used in this study had a specific activity of 195 U/μg.

### Purification of LTA

LTA purification was done essentially following our published protocol[77]. Bacterial pellets were suspended in citrate buffer (50 mM, pH 4.7) and disrupted three times by French press (Constant Cell Disruption System, Serial No. 1020) at 10 °C at a pressure of 20 kPSI (*Spn* D39) or 40 kPSI (*Sa*), respectively. SDS was added to a final

concentration of 4% to the combined supernatants. The solution was incubated for 30 min at 100 °C and was stirred at RT overnight. The solution was centrifuged at 30,000 × g for 15 min at 4 °C. The pellet was washed four times with citrate buffer using the centrifugation conditions as above. The combined LTA-containing supernatants were lyophilized. The resulting solid was washed five times with ethanol (centrifugation: 20 min, 20 °C, 10,650 × g) to remove SDS and lyophilized. The resulting pellet was suspended in citrate buffer and extracted with an equal volume of butan-1-ol (Merck) at RT under vigorous stirring. The phases were separated by centrifugation at 2100 × g for 15 min at 4 °C. The aqueous phase (containing LTA) was collected, and the extraction procedure was repeated twice with the organic phase plus interphase. The combined aqueous phases were lyophilized and subsequently dialyzed for 5 days at 4 °C against 50 mM ammonium acetate buffer (pH 4.7; 3.5 kDa cutoff membrane); the buffer was changed every 24 h. The resulting crude LTA was purified further by hydrophobic interaction chromatography (HIC) performed on a HiPrep Octyl-Sepharose column (GE Healthcare; 16 × 100 mm, bed volume 20 ml). The LTA-containing pellet was dissolved in the HIC starting buffer at a concentration of 15 mg ml⁻¹ and purified by HIC using a linear gradient from 15 to 60% propan-1-ol (Roth) in 0.1 M ammonium acetate (pH 4.7). The LTA-containing fractions were combined, lyophilized, and washed with water upon freeze drying to remove residual buffer. The following bacterial strains were used for LTA purification: *Spn* D39Δ*cps*, *S. aureus* 113. The *S. aureus* strain was kindly provided by F. Götz, Tübingen, Germany.

### Culture and transfection of BEAS-2B cells

The human bronchial epithelial cell line BEAS-2B (CRL-9609, ATCC, discontinued or Merck, 95102433) was routinely cultured in BEGM (Lonza, CC-3071) according to ATCC protocol without use of fibronectin coating. Cells were seeded at 10⁵ cells/cm2 and cultivated to 80% confluence (usually overnight). Medium was exchanged, followed immediately by infection or stimulation (see below). Transfection experiments were performed with 20 nM scramble control (4390843, Ambion), siNAMPT (Dharmacon, L-004581-00-0005) or siNMNAT1 (Dharmacon, L-008951-00-0005) in Lipofectamin RNAiMAX (diluted 1:100, Invitrogen). Lipofectamin and siRNA were mixed in OptiMem medium (ThermoFisher, 51985034) and incubated for 10 min at room temperature. From the transfection mix, 100 µl was used to transfect approximately 2 × 10⁵ cells in 400 µl BEGM (total volume 500 µl). Transfection was performed when seeding cells. After transfection, cells were cultivated for 48 h and used for infection experiments as described.

### Culture and infection of primary human bronchial epithelial cells

Primary human bronchial epithelial cells (HBEC) from healthy donors were obtained and cultivated in an air-liquid interface system[78]. 6 × 10⁴ cells/cm² were seeded onto collagenized (Type 1 Collagen, Merck, CC050) Costar Transwell Permeable supports (3460, Corning) in Airway Epithelial Growth Medium (AEGM, Promocell, C-21160)). After reaching confluence, they were airlifted by aspiration of apical medium. Medium in the basolateral chamber was changed to differentiation medium (DMEM (ThermoFisher, 31966021)/AEGM (1:1) supplemented with 0.1 ng/mL retinoic acid (Merck) and penicillin/streptomycin). Cells were cultivated for 28 days and barrier formation was verified by measuring transepithelial electrical resistance (TEER). All donors developed a TEER > 880 Ω/cm². Cells were used at passages 3 or 5.

Basolateral medium was exchanged for differentiation medium without antibiotics on differentiation day 26, and again immediately before infection on day 29. To assess the effect of NMN on *Spn* growth, *Spn* D39 were added apically at MOI 20 in 50 µl PBS. After one hour of incubation, cells were washed with 400 µl PBS. NMN 500 µM was added basolateral and cells were incubated for 16 h. For gene expression analysis, cells were apically infected with *Spn* D39 at MOI 5 or 20 in 10 µl PBS without washing for 16 h.

### Preparation and infection of human lung tissue explants

For preparation of tissue explants, tumor-distant, macroscopically tumor-free human lung tissue was obtained from tumor resections of bronchial carcinoma patients. For infection with *S. pneumoniae*, lung tissue was stamped into cylinders (~8 × 8 mm) and weighed. Specimens were incubated for 24 h in RPMI 1640 with 10% (vol/vol) heat-inactivated fetal calf serum (except for bacterial growth) prior to infection[70]. 200 µL prepared control or bacteria-containing infection medium (10⁶ CFU/160 mg tissue) was injected per 100 mg tissue and incubated for 12 h. Tissue was collected at the University Medical Centre Marburg by Andreas Kirschbaum in agreement with local ethics regulations (Marburg 161/17) after obtaining written informed consent from patients.

### mRNA expression analysis

BEAS-2B cells were lysed in TRIzol™ (ThermoFisher, 15596026) and RNA was isolated by phenol-chloroform extraction. Explorative transcriptomic analysis was performed using DNA microarrays (Human G3 v2 Kit, 8x60k, Agilent technologies). Microarray data are published in GEO under the accession GSE195778. In brief, purified total RNA was amplified using the Agilent Low Input QuickAmp kit, 200 ng labelled aRNA was hybridized following the Agilent protocol. Slides were scanned using the Innoscan 900 scanner (Innopsys, Carbononne France) at 2 µp/pixel and images were analyzed with Mapix 6.5.0. Raw data was processed in R using limma[79]. Background correction was performed with the normexp model and spot intensities were quantile-normalized between arrays. For quantitative PCR analysis, isolated mRNA was reverse transcribed to cDNA using the High-Capacity cDNA Reverse Transcription Kit (Thermo Fisher Scientific) following manufacturer's protocol and detected using the Fast SYBR Green Master Mix (Thermo Fisher Scientific) according to manufacturer's protocol. The following custom-made primers were used for qPCR: IL8: fwd: 5′-ACTGAGAGTGATTGAGAGTGGAC-3′, rev: 5′-AACCCTCTGCACCCAGT TTTC-3′; NAMPT: fwd: 5′- GGTTACAAGTTGCTGCCACC-3′, rev: 5′-AGCAAACCTCCACCAGAACC; NMNAT1: fwd: 5′-GTGATCTCCGGTAGC ACTCG-3′, rev: 5′-CTTGGCCAGCTCAAACAACC-3′; NNMT: fwd: 5′-TAA GGAGATCGTCGTCACTG-3′, rev: 5′-CTGCTTGACCGCCTGTCTC-3′; RPS18: fwd: 5′-GCGGCGGAAAATAGCCTTTG-3′, rev: 5′-GATCACAC GTTCCACCTCATC-3′. All samples were processed on a Quantstudio qPCR device (Life Technologies) with QuantStudio Real-Time PCR software (ThermoFisher, v1.3). Gene expression was calculated as ΔΔCT values and normalized towards mock-treated controls.

### Proteomic analysis

For SILAC (stable isotope labelling with amino acids in cell culture) standard generation, BEAS-2B cells were cultivated in DMEM with 2% FCS with heavy isotopes of lysine and arginine (EurisoTop) for at least three passages. For sample generation, cells were cultivated in DMEM with 2% FCS, without addition of labelled amino acids and control-treated or infected with *Spn* D39 MOI 0.5 for 16 h as described above. Cells were then washed with PBS and harvested in solubilisation buffer (8 M urea, 2 M thiourea in H₂O) and proteins were extracted by 5 freeze-thaw-circles combines with ultrasonification (3x3 seconds; SonoPuls, Bandelin electronic, Germany)[80]. Each control or infected sample was mixed 1:1 with the marked standard and analysed by 1D gel electrophoresis (NuPAGE 4–12% Acrylamide Bis-Tris Medi Gel, Novex, Life Technologies) according to the manufacturer's protocol. All bands were separately extracted from the gel and digested with 10 µg/ml trypsin, following peptide extraction and C₁₈ purification (Merck Millipore). Five fractions of each sample were separated by nanoLC (Dionex UltiMate 3000, Dionex/ThermoFisher Scientific), ionized by

TriVersa NanoMate (Advion, Ltd.) and measured by mass spectrometry (Q Exactive™ Hybrid-Quadrupole-Orbitrap, Thermo Fisher Scientific). By using Proteome Discoverer™ 1.4 (Thermo Fisher Scientific), detected peptides were mapped to the Uniprot protein database limited to human entries. Per sample and identified protein, a light-to-heavy (i.e., sample-to-standard) ratio was calculated. These ratios were used to further calculate log2 fold-changes (infected vs control) and q-values by moderated t-test (data analysis described in[81]). The mass spectrometry proteomics data have been deposited to the ProteomeXchange Consortium via the PRIDE partner repository[82] with the dataset identifier PXD039059. Details on data acquisition are outlined in Table S1.

## Metabolite analysis

After 16 h of infection with *Spn* D39 MOI 1, infected BEAS-2B cells or uninfected controls were washed with 0.9%, 37 °C NaCl solution. Cell culture plates were placed on ice and cells lysed in ice-cold Tris-EDTA/MeOH buffer (100 mM Tris, 1 mM EDTA in *A.dest*, 1:1 mixture with 98% MeOH). An equal volume of ice-cold chloroform was added, and samples were incubated for 30 min at 4 °C prior to centrifugation (10,000 g, −10 °C, 10 min). Supernatants were filtered (Minisart RC4, 0.2 μM, Sartorius), snap-frozen in liquid nitrogen and stored at −80 °C until analysis. Tris-EDTA/MeOH and chloroform were gassed with $N_2$ immediately before use to prevent oxidation of metabolites. Quantitative determination of intracellular metabolites was performed using LC-MS/MS. The chromatographic separation was performed on an Agilent Infinity II 1290 HPLC system using a ZicpHILIC SeQuant column (150 × 2.1 mm, 5 μm particle size, 100 Å pore size) connected to a guard column of the same specificity (Merck) at a constant flow rate of 0.35 ml/min, with mobile phase A being 10 mM ammonium hydroxide in water adjusted to a pH of 9.8, and eluent B being acetonitrile (Honeywell) at 30 °C. The injection volume was 2 μl. The mobile phase profile consisted of the following steps and linear gradients: 0–7 min from 90 to 55% B; 7–10 min constant at 55% B; 10–10.1 min from 55 to 90% B; 10.1–12.5 min constant at 90% B. An Agilent 6495 ion funnel mass spectrometer was used in positive mode with an electrospray ionization source and the following conditions: ESI spray voltage 1500 V, nozzle voltage 500 V, sheath gas 400 °C at 12 l/min, nebulizer pressure 30 psig and drying gas 250 °C at 13 l/min. Compounds were identified based on their mass transition and retention time compared to standards. Chromatograms were integrated using MassHunter software (Agilent, Santa Clara, CA, USA). Absolute concentrations were calculated based on an external calibration curve prepared in sample matrix.

## ELISA and Western Blot

Concentration of IL-8 in the cell supernatant was analysed with a commercial ELISA kit (OptEIA, BD Biosciences) according to the manufacturer's instructions. For Western Blotting, cells were harvested in RIPA buffer and cell debris removed by centrifugation (8000 g, RT). SDS-PAGE (10% polyacrylamide) and wet blotting to a nitrocellulose membrane was performed with 25 μg protein as determined by BCA assay. For NAMPT detection, rabbit anti-human NAMPT (dilution 1:500; Thermo Fisher Scientific, PA1-1045) and anti-rabbit-HRP (dilution 1:1,000, NEB, 5127 S) were used. Actin detection was performed using the Goat anti-human actin (dilution 1:1,000, SantaCruz, sc-1616) and anti-goat HRP (dilution 1:5,000, SantaCruz, sc-2020). Turnover of ECL substrate (GE Healthcare, 28980926) was detected on a chemoluminescence imager (INTAS Science Imaging Instruments).

## NAD+/NADH measurement

Total extracellular, intracellular and intrabacterial concentration of NAD (NAD⁺ and NADH) was determined using a commercial colorimetric assay (Abcam, ab65348) according to manufacturer's instructions on a Tecan Infinite M200 PRO (ThermoFisher). Approximately $2 \times 10^5$ BEAS-2B cells were infected with *Spn* D39 for 16 h as described above. After

infection, cells were lysed and NAD⁺/NADH concentrations were determined. Cell counts of infected and uninfected samples were determined for normalization. To calculate intracellular concentrations, a BEAS-2B volume of 2.2 pl/cell was assumed based on literature[83].

## Reporting summary

Further information on research design is available in the Nature Portfolio Reporting Summary linked to this article.

## Data availability

The mRNA microarray data generated in this study have been deposited in the GEO accession viewer under accession code GSE195778. The proteome data generated in this study have been deposited in the PRIDE database under the accession number PXD03905. Bacterial sequencing data generated in this study have been deposited in the BioStudies database under the accession number E-MTAB-12644. Source data are provided with this manuscript. Biological material and bacterial strains are available from Bernd Schmeck upon reasonable request. Source data are provided with this paper.

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

## Acknowledgements

The authors wish to thank Manuela Gesell Salazar for conducting the mass spectrometric measurements and Kerstin Hoffmann and Peter Claus as well as the team of the Microscopy CORE Lab, Maastricht for their excellent technical assistance. This work has been funded in part by the Bundesministerium für Bildung und Forschung (Federal Ministry of Education and Research: PermedCOPD – FKZ 01EK2203A; ERACo-SysMed2 – SysMed-COPD – FKZ 031L0140; e:Med CAPSYS – FKZ 01ZX1604E), the Deutsche Forschungsgemeinschaft (SFB/TR-84 TP C01 to B.S.; SFB/TR-84 TP A4 to M.A.; HA 3125/5-2 to S.H.; GI 979/1-2 to N.G.), the von-Behring-Röntgen-Stiftung (66-LV07) to B.S., and the Hessisches Ministerium für Wissenschaft und Kunst (LOEWE Diffusible Signals LOEWE-Schwerpunkt Diffusible Signals) to B.S., A.L.J., A.B. and E.M.E. Illustrations for Fig. 2A and Fig. 2D were created using biorender.com.

## Author contributions

Conceptualization: B.K., A.W., E.V., B.B., B.S.; Methodology: B.K., A.W., K.S., J.S., K.K., U.V., E.V., B.B., B.S.; Formal analysis: B.K., W.B., J.W., N.P., J.S., B.B.; Investigation: B.K., A.W., K.S., S.B., K.K., I.B., J.S.; Resources: A.K., M.A.M., S.H., B.A., N.G.; Writing – Original Draft: B.K., B.B.; Writing: Review and Editing. B.K., A.W., W.B., N.P., K.S., J.W., J.S., E.M.E., A.K., K.L., A.J., A.B., U.V., E.V., B.B., B.S.; Visualization: B.K., E.M.E., B.B.; Supervision: A.B., U.V., E.V., B.B., B.S.; Funding acquisition: B.S.

## Funding

## Competing interests

The authors declare no competing interests.

## Additional information

[1]Institute for Lung Research, Universities of Giessen and Marburg Lung Center (UGMLC), German Center for Lung Research (DZL), Philipps-Universität Marburg, Marburg, Germany. [2]Core Facility for Metabolomics and Small Molecule Mass Spectrometry, Max Planck Institute for Terrestrial Microbiology, Marburg, Germany. [3]Department of Functional Genomics, Interfaculty Institute for Genetics and Functional Genomics, University Medicine Greifswald, Greifswald, Germany. [4]Institute for Lung Health (ILH), Giessen, Germany. [5]Universities of Giessen and Marburg Lung Center (UGMLC), Justus-Liebig-Universität Giessen, German Center for Lung Research (DZL), Giessen, Germany. [6]Center for Synthetic Microbiology (SYNMIKRO), Philipps-Universität Marburg, Marburg, Germany. [7]Microscopy CORE Lab, Maastricht Multimodal Molecular Imaging Institute (M4I), Maastricht University, Universiteitssingel 50, 6229 ER Maastricht, The Netherlands. [8]Department of Biology, Philipps-Universität Marburg, Marburg, Germany. [9]Department of Microbiology and Immunology, Faculty of Pharmacy, Zagazig University, Zagazig, Egypt. [10]Core Facility Flow Cytometry – Bacterial Vesicles, Philipps-Universität Marburg, Marburg, Germany. [11]Department of Visceral, Thoracic and Vascular Surgery, University Hospital Gießen and Marburg (UKGM), Marburg, Germany. [12]Department of Molecular Genetics and Infection Biology, Interfaculty Institute for Genetics and Functional Genomics, Center for Functional Genomics of Microbes, University of Greifswald, Greifswald, Germany. [13]Division of Bioanalytical Chemistry, Priority Area Infections, Research Center Borstel, Leibniz Lung Center, Borstel, Germany. [14]Institute for Medical Microbiology, Justus-Liebig Universität Giessen, Giessen, Germany. [15]University Eye Clinic Maastricht, Maastricht University Medical Center (MUMC+), School for Mental Health and Neuroscience, Maastricht University, P. Debyelaan 25, 6229 HX Maastricht, The Netherlands. [16]Department of Medicine, Pulmonary and Critical Care Medicine, University Medical Center Marburg, Philipps-Universität Marburg, Marburg, Germany. [17]Member of the German Center for Infectious Disease Research (DZIF), Marburg, Germany. [18]These authors contributed equally: Birke J. Benedikter, Bernd Schmeck. ✉e-mail: b.benedikter@maastrichtuniversity.nl; schmeck@staff.uni-marburg.de

