## [Peer Review File · Nature Communications]

NAD⁺ metabolism is a key modulator of bacterial respiratory epithelial infectionsREVIEWER COMMENTS

Reviewer #1 (Remarks to the Author):

The work of Klabunde and coworkers is noteworthy in showing how Spn infections disturb the NAD system both from the point of view of host innate immune response and bacterial bioenergetics. The article will deserve publication after the following matters are addressed.

1) The simultaneous induction of NNMT and NAMPT would be expected to have complex interactions with host NAD metabolism because NNMT is dumping Nam into a nonsalvageable form (MeNAM) while NAMPT is driving Nam salvage. The authors should think further about this and comment.

2) NAMPT might be the major salvage pathway but in truth it depends on the cell type and what is available to it. Bogdan and Brenner reviewed this in *Ann Rev Nutr* in 2008.

3) The interaction between host NAD metabolism, innate immunity and viral infection has been extensively studied and there are some common features to what is seen here. Authors are referred to Heer et al on Coronavirus (*J Biol Chem* 2020) and my 2022 review in *Nature Metabolism*.

4) Adding NAD⁺ or NMN to cell cultures worked in this paper but do not constitute particularly sound methods as NAD⁺ contains two phosphates, NMN contains a phosphate and it has been clear since the work of Gertrude Elion that nucleotides don't get into cells intact. A quick examination of the bacterial genome will allow Klabunde to determine if Spn has a NR kinase in which case NR can be used for both experiments. The other options are nicotinamide and nicotinic acid. Work in the olden days showed that NAD and NMN both release NR for growth of *Haemophilus influenzae*. They could expand on this work in the future by knocking out the Spn NR kinase to look at failure to respond to both NMN and NR. For the purposes of this paper, it would suffice to show that NR carries the activity of NAD⁺ and NMN in the existing figures.

Charles Brenner

Reviewer #2 (Remarks to the Author):

This is an interesting manuscript investigating the role of NAD⁺ and the NAD⁺ salvage pathway in the respiratory epithelium during pneumococcal infection. The authors used a multi-omics approach to identify pathways differentially expressed within epithelial cells during pneumococcal infection. Additionally, the authors provide evidence that pneumococcus interferes with this pathway via its cytolytic toxin, pneumolysin. The manuscript is well-written and the conclusions are supported by the data presented. Specific points are provided below.

Major points:

Some of the data presented in Supplementary figures are not discussed within the manuscript. For instance, the data from Figure S1 is not even mentioned or discussed. The only reference to this figure is in Line 77 when the authors say they established an in vitro infection model. If the data is not going to be discussed, it should be removed.

The authors don't seem to address why they initially identified upregulation of both NAMPT and NMNAT1 in BEAS-2B cells and in other cells/infections identified NMNAT1 to be downregulated. It is assumed that pneumolysin would've been produced in the initial BEAS-2B experiment.

Reviewer #3 (Remarks to the Author):

This study used -omics approaches to identify epithelial cell pathways involved in longer-term *Streptococcus pneumoniae* lung infection. Proteomics following lung cell infection identified 2 upregulated enzymes involved in NAD⁺ metabolism, and the results were validated by mRNA microarray, qPCR, and immunoblot. Metabolite analysis showed a decrease in NAD⁺ and upstream precursors (NCA, NAAD, NAM, and NMN) as well as enzymes (NAMPT, NNMT, and QAPRT) following infection, indicating a dysregulation of the Preiss-Handler and NAD⁺ Salvage pathways. Results from the lung cells were then confirmed for a primary human lung cell line, explanted human lung tissue, and publicly available data from mouse models of lung infection. siRNA knockdowns of NAMPT and NMNAT1 resulted in reduction of NAD⁺ and increased bacterial load, and addition of NAD⁺ decreased bacterial load (albeit a modest reduction). Boosting lung cells with NMN increased NAD⁺ production and reduced bacterial load, and additional experiments modulating the pathway components further verified this finding. NAD⁺ was determined to be a key component of the host lung epithelial cell response to *S. pneumoniae* in that it decreased bacterial fitness. A role for pneumolysin in regulation of NAD⁺ was determined, and induction of NAD⁺ resistance by *S. pneumoniae* was associated with capsule locus mutation resulting in loss of capsule. Overall, this study possessed rigor and characterized a mechanism by which host cell NAD⁺ metabolism influences pneumococcal fitness, and likewise how *S. pneumoniae* components influence the host cell during lung infection. The Discussion is nicely done with thought given to limitations.

The authors state that upregulation of NAMPT appears to be a general pro-inflammatory response based on its upregulation following infection with 2 additional pathogens (Figure S5 and discussion). For *S. pneumoniae* and *S. agalactiae*, this upregulation could be due at least in part by LTA since the purified *S. aureus* LTA control produced similar results (Figure S3). It would be interesting for the authors to speculate what alternative Gram-negative component of *H. influenzae* might be inducing upregulation, if relevant.

In analyzing whether hydrogen peroxide or pneumolysin production were involved in NAMPT/NMNAT1 expression, it was determined that hydrogen peroxide played no role whereas pneumolysin did. However, instead of using an *spxB* mutant of *S. pneumoniae* to analyze the effect of hydrogen peroxide, 80 μ M H₂O₂ was added to cells (Figure S6). Heat or UV-killed *S. pneumoniae* had no effect and therefore it was hypothesized that repression was actively done by the bacteria; therefore, it does not make sense why the same approach for pneumolysin (using a knockout strain) was not also used for H₂O₂.

Another finding that is disparate is that LTA alone caused upregulation of NAMPT (Figure S3), but killed *S. pneumoniae* did not (Figure S6). What is the explanation for these disparate results? LTA from *S. aureus* was used in the experiments, which is of a different chemical composition than the LTA from *S. pneumoniae*. LTA from *S. pneumoniae* should be extracted and tested for the ability to upregulate NAMPT.

Line 475: The "purification of pneumolysin" section of the methods should include more details.

RESPONSE TO REVIEWERS

Reviewer #1 (Remarks to the Author):

The work of Klabunde and coworkers is noteworthy in showing how Spn infections disturb the NAD system both from the point of view of host innate immune response and bacterial bioenergetics. The article will deserve publication after the following matters are addressed.

We thank Prof. Brenner for his thorough review and positive assessment of our work. In the manuscript, we marked changes in response to Prof. Brenner's review in yellow.

1) The simultaneous induction of NNMT and NAMPT would be expected to have complex interactions with host NAD metabolism because NNMT is dumping Nam into a nonsalvageable form (MeNAM) while NAMPT is driving Nam salvage. The authors should think further about this and comment.

We agree that the upregulation of NNMT is noteworthy and, albeit not the focus of this manuscript, deserves more prominent mentioning. We expanded the discussion accordingly (Line 286-287, 291-292, 420-427).

2) NAMPT might be the major salvage pathway but in truth it depends on the cell type and what is available to it. Bogan and Brenner reviewed this in Ann Rev Nutr in 2008.

We added this information to the introduction (line 65-66).

3) The interaction between host NAD metabolism, innate immunity and viral infection has been extensively studied and there are some common features to what is seen here. Authors are referred to Heer et al on Coronavirus (J Biol Chem 2020) and my 2022 review in Nature Metabolism.

We feel that the results mentioned connect well to the results we obtained and included them in the discussion (Line 299-300, 322-325).

4) Adding NAD⁺ or NMN to cell cultures worked in this paper but do not constitute particularly sound methods as NAD⁺ contains two phosphates, NMN contains a phosphate and it has been clear since the work of Gertrude Elion that nucleotides do not get into cells intact. A quick examination of the bacterial genome will allow Klabunde to determine if Spn has a NR kinase in which case NR can be used for both experiments. The other options are nicotinamide and nicotinic acid. Work in the olden days showed that NAD and NMN both release NR for growth of Haemophilus influenza. They could expand on this work in the future by knocking out the Spn NR kinase to look at failure to respond to both NMN and NR. For the purposes of this paper, it would suffice to show that NR carries the activity of NAD⁺ and NMN in the existing figures.
Charles Brenner

In *Spn*, PnuC was reported to be the main transporter involved in NR uptake (Johnson, 2015). We repeated the experiments in Figure 4 using NR and NAM to increase NAD⁺ levels in the eukaryotic cell during infection. The results we obtained confirm our previous results: NMN, NAM and NR treatment of BEAS-2B cells and human bronchial epithelial cells increased intracellular NAD⁺/NADH levels and reduced bacterial counts after infection. In contrast, in absence of cells, NR, NAM and NMN treatment exhibited minor or no effect on bacterial counts.

Additionally, bacterial counts were partially rescued by inhibition of NAD⁺ biosynthesis by siNMNAT1 knockdown prior to NR/NAM/NMN treatment. This supports our assumption that host cell mediated NAD⁺ biosynthesis is necessary for NR, NMN and NAM to exhibit antibacterial activity. Our newly obtained results therefore strengthen our conclusion that NAD⁺ is the main carrier of antibacterial activity and its precursors act antibacterial by increasing NAD⁺ production. We prepared a new Figure 4 to present the results obtained from NR treatment. We further prepared a new supplemental Figure S3, which includes results obtained by NAM and NMN treatment of *Spn* in presence and absence of host cells and with and without siNMNAT1 knockdown. The description of results and the discussion were adjusted accordingly (line 149-171; 325-343). We furthermore expanded the discussion by a paragraph discussing the potential mechanisms of antibacterial activity of NAD⁺ (line 391-408).

Reviewer #2 (Remarks to the Author):

This is an interesting manuscript investigating the role of NAD⁺ and the NAD⁺ salvage pathway in the respiratory epithelium during pneumococcal infection. [...] The manuscript is well-written and the conclusions are supported by the data presented. Specific points are provided below.

In the manuscript, we marked changes in response to reviewer 2 in light blue.

Major points:

Some of the data presented in Supplementary figures are not discussed within the manuscript. For instance, the data from Figure S1 is not even mentioned or discussed. The only reference to this figure is in Line 77 when the authors say they established an in vitro infection model. If the data is not going to be discussed, it should be removed.

To improve clarity of the manuscript, we removed the former supplemental figures S1 (Establishment of infection) and S3 (Confirmation of Omics-data) which were only mentioned in the results with one sentence each and adjusted the manuscript accordingly.

The authors don't seem to address why they initially identified upregulation of both NAMPT and NMNAT1 in BEAS-2B cells and in other cells/infections identified NMNAT1 to be downregulated. It is assumed that pneumolysin would've been produced in the initial BEAS-2B experiment.

We apologise for any confusion regarding the identified gene regulations. In our study, upon pneumococcal infection, expression of NMNAT1 was downregulated in all models tested. In contrast, NAMPT and NNMT expression was upregulated upon infection. We were unable to find a sentence where we accidentally stated otherwise. Should we have made such a mistake, we would kindly ask the reviewer to point out the exact line so that we can correct it. We also slightly modified the description of the results presented in Figure 1 to improve clarity (Line 89).

Reviewer #3 (Remarks to the Author):

This study used -omics approaches to identify epithelial cell pathways involved in longer-term *Streptococcus pneumoniae* lung infection. Proteomics following lung cell infection identified 2 upregulated enzymes involved in NAD⁺ metabolism, and the results were validated by mRNA microarray, qPCR, and immunoblot. Metabolite analysis showed a decrease in NAD⁺ and upstream precursors (NCA, NAAD, NAM, and NMN) as well as enzymes (NAMPT, NNMT, and QAPRT) following infection, indicating a dysregulation of the Preiss-Handler and NAD⁺ Salvage pathways. Results from the lung cells were then confirmed for a primary human lung cell line, explanted human lung tissue, and publicly available data from mouse models of lung infection. siRNA knockdowns of NAMPT and NMNAT1 resulted in reduction of NAD⁺ and increased bacterial load, and addition of NAD⁺ decreased bacterial load (albeit a modest reduction). Boosting lung cells with NMN increased NAD⁺ production and reduced bacterial load, and additional experiments modulating the pathway components further verified this finding. NAD⁺ was determined to be a key component of the host lung epithelial cell response to *S. pneumoniae* in that it decreased bacterial fitness. A role for pneumolysin in regulation of NAD⁺ was determined, and induction of NAD⁺ resistance by *S. pneumoniae* was associated

with capsule locus mutation resulting in loss of capsule. Overall, this study possessed rigor and characterized a mechanism by which host cell NAD⁺ metabolism influences pneumococcal fitness, and likewise how *S. pneumoniae* components influence the host cell during lung infection. The Discussion is nicely done with thought given to limitations.

In the manuscript, we marked changes in response to reviewer 3 in green.

The authors state that upregulation of NAMPT appears to be a general pro-inflammatory response based on its upregulation following infection with 2 additional pathogens (Figure S5 and discussion). For *S. pneumoniae* and *S. agalacticae*, this upregulation could be due at least in part by LTA since the purified *S. aureus* LTA control produced similar results (Figure S3). It would be interesting for the authors to speculate what alternative Gram-negative component of *H. influenzae* might be inducing upregulation, if relevant.

The induction of a seemingly TLR2-dependent regulatory mechanism by a Gram-negative pathogen is indeed an interesting observation. We found in literature that this effect might be mediated by porin of *H. influenzae* and included this information in the discussion (line 305-307)

In analyzing whether hydrogen peroxide or pneumolysin production were involved in NAMPT NMNAT1 expression, it was determined that hydrogen peroxide played no role whereas pneumolysin did. However, instead of using an *spxB* mutant of *S. pneumoniae* to analyze the effect of hydrogen peroxide, 80 μ M H₂O₂ was added to cells (Figure S6). Heat or UV-killed *S. pneumoniae* had no effect and therefore it was hypothesized that repression was actively done by the bacteria; therefore, it does not make sense why the same approach for pneumolysin (using a knockout strain) was not also used for H₂O₂.

To ensure comparability of results across the whole manuscript, we infected BEAS-2B cells with a Δ *spxB* mutant of *SpnD39* and analysed expression of NAMPT and NMNAT1. We furthermore infected BEAS-2B cells with *SpnD39* Δ *cps* to investigate the main virulence factors of *Spn* in a comparable manner. The results confirm that neither pneumococcal H₂O₂ production nor the capsule are involved in regulation of host cell NAD⁺ biosynthesis. The results are included in Figure S5 (former S6).

Another finding that is disparate is that LTA alone caused upregulation of NAMPT (Figure S3), but killed *S. pneumoniae* did not (Figure S6). What is the explanation for these disparate results? LTA from *S. aureus* was used in the experiments, which is of a different chemical composition than the LTA from *S. pneumoniae*. LTA from *S. pneumoniae* should be extracted and tested for the ability to upregulate NAMPT.

We obtained LTA from Prof. Sven Hammerschmidt/Dr. Nicolas Gisch and stimulated BEAS-2B cells with LTA isolated from *Spn* and *S. aureus* and determined the expression of NAMPT and NMNAT1. The results confirm the hypothesis of the reviewer: LTA isolated from *S. aureus* caused a significantly stronger induction of NAMPT than LTA isolated from *Spn*. We included this information in Figure S5.

Line 475: The "purification of pneumolysin" section of the methods should include more details.

We expanded the paragraph by adding details regarding protein purification and purity assessment.

Sincerely,

Björn Klabunde

Bernd Schmeck

REVIEWERS' COMMENTS

Reviewer #1 (Remarks to the Author):

the authors have effectively revised their interesting paper on NAD metabolism as a modulator of bacterial infection of respiratory epithelial cells...CB

Reviewer #3 (Remarks to the Author):

The authors responded to the first round of review with additional experiments and text. One small item that was overlooked is that they should reference prior publications that generated the cps and spxB mutants, or briefly describe their construction, in the methods.

Marburg, 16.08.2023

RESPONSE TO REVIEWERS

Reviewer #3 (Remarks to the Author):

The authors responded to the first round of review with additional experiments and text. One small item that was overlooked is that they should reference prior publications that generated the cps and spxB mutants, or briefly describe their construction, in the methods.

We added references to the initial description of bacterial strains used (page 18, marked yellow).

Sincerely,

Björn Klabunde

Bernd Schmeck